# MICU2 up-regulation enhances tumor aggressiveness and metabolic reprogramming during colorectal cancer development

Alison Robert[1ʘ], David Crottès[1ʘ], Jérôme Bourgeais[1], Naig Gueguen[2], Arnaud Chevrollier[2], Jean-François Dumas[1], Stéphane Servais[1], Isabelle Domingo[1], Stéphanie Chadet[1], Julien Sobilo[3], Olivier Hérault[1], Thierry Lecomte[1], Christophe Vandier[1], William Raoul[1], Maxime Guéguinou[1]*

1 UMR Inserm 1069 N2COx « Niche, Nutrition, Cancer et métabolisme Oxydatif », Tours University, Tours, France, 2 CNRS UMR 6015, Inserm U1083 MITOVASC, MitoLab team, Angers University, Angers, France, 3 PHENOMIN-TAAM-CNRS UPS44, Orléans, France

ʘ These authors contributed equally to this work.
* maxime.gueguinou@univ-tours.fr

## Abstract

The mitochondrial $Ca^{2+}$ uniporter (MCU) plays crucial role in intramitochondrial $Ca^{2+}$ uptake, allowing $Ca^{2+}$-dependent activation of oxidative metabolism. In recent decades, the role of MCU pore-forming proteins has been highlighted in cancer. However, the contribution of MCU-associated regulatory proteins mitochondrial calcium uptake 1 and 2 (MICU1 and MICU2) to pathophysiological conditions has been poorly investigated. Here, we describe the role of MICU2 in cell proliferation and invasion using in vitro and in vivo models of human colorectal cancer (CRC). Transcriptomic analysis demonstrated an increase in MICU2 expression and the MICU2/MICU1 ratio in advanced CRC and CRC-derived metastases. We report that expression of MICU2 is necessary for mitochondrial $Ca^{2+}$ uptake and quality of the mitochondrial network. Our data reveal the interplay between MICU2 and MICU1 in the metabolic flexibility between anaerobic glycolysis and OXPHOS. Overall, our study sheds light on the potential role of the MICUs in diseases associated with metabolic reprogramming.

## Introduction

Intracellular calcium ($Ca^{2+}$), which is a ubiquitous second messenger, regulates a multitude and various cellular processes. Deregulation of $Ca^{2+}$ homeostasis and $Ca^{2+}$-dependent signaling pathways have been described to be associated with each of the "cancer hallmarks" [1,2]. Mitochondria are a central player in maintaining cellular $Ca^{2+}$ homeostasis by uptaking $Ca^{2+}$ from cytoplasm.

Fine regulation of $Ca^{2+}$ fluxes in mitochondria is required to support functions linked to metabolism (ATP production), redox homeostasis, and proliferation/apoptosis balance.

**Data Availability Statement:** All relevant data are within the paper and its Supporting Information files.

**Funding:** This project was supported by French departmental committees of Ligue Contre le Cancer "Grand-Ouest": 16 (Charente), 37 (Indre-et-Loire), 49 (Maine-et-Loire), 72 (Sarthe) and 85 (Vendée) (to MG and WR), Inserm (to MG), Fondation ARC (to MG), Labex MabImprove (ANR-10-LABX-53-01) (to MG), INCa (INCa-PLBIO 18-151 and 18-145) (to MG, CV and WR), Université de Tours and Région Centre-Val de Loire APR-IA CAMITHERAPAL (to MG). AR received salary from INSERM / Région Centre Val-de-Loire (3 year doctoral grant). DC received one-year salary from Le Studium Loire Valley Institute for Advanced Studies (Smart Loire Valley Fellowship). The funders had no role in study design, data collection and analysis, decision to publish, or preparation of the manuscript.

**Competing interests:** The authors have declared that no competing interests exist

**Abbreviations:** ADP, adenosine diphosphate; BN-PAGE, blue native polyacrylamide gel electrophoresis; BSA, bovine serum albumin; CRC, colorectal cancer; CS, citrate synthase; DEA, differential expression analysis; DMEM, Dulbecco's Modified Eagle Medium; ER, endoplasmic reticulum; FBS, fetal bovine serum; GSEA, gene set enrichment analysis; IMM, inner mitochondrial membrane; KO, knockout; LDH, lactate dehydrogenase; LPA, lysophosphatidic acid; MCU, mitochondrial calcium uniporter; OCR, oxygen consumption rate; PBS, phosphate-buffered saline; PCR, polymerase chain reaction; PSS, physiological saline solution; ROS, reactive oxygen species; SOCE, store-operated calcium entry; TCA, tricarboxylic acid; VDAC, voltage-dependent anion-selective channel.

Mounting evidence indicates that metabolic transformation of tumor cell is the main limitation of cancer treatment, which is highly related to the resistance to therapeutic drugs [3]. Due to its role in energetic metabolism by activating pyruvate dehydrogenase, isocitrate dehydrogenase, and α-ketoglutarate dehydrogenase, or respiratory complex activity, mitochondrial $Ca^{2+}$ (mito$Ca^{2+}$) plays a pivotal role in cancer progression [4–7].

To enter in the mitochondrial matrix, $Ca^{2+}$ first passes through the outer membrane of the mitochondrion via the voltage-dependent anion-selective channel (VDAC), then through the inner mitochondrial membrane (IMM) via the mitochondrial calcium uniporter (MCU), a highly selective $Ca^{2+}$ channel. MCU forms a large multi-molecular complex composed of different proteins. This channel, alongside the MCU regulator EMRE, is located in the IMM. Proteins regulating mito$Ca^{2+}$ uptake such as mitochondrial calcium uptake 1 (MICU1), MICU2, and MICU3 reside in the intermembrane space [8,9]. The regulation of MCU activity by MICU1 and MICU2 involves a gating mechanism. MCU gating and the role of MICU subunits in potentiation are dependent on the $Ca^{2+}$ content and the MCU current. Recently, Garg and colleagues [10] demonstrated that elevation of cytosolic $[Ca^{2+}]$ increases $Ca^{2+}$ binding to the EF hands of MICU subunits. MICUs thereby increase the open probability of MCU, enhancing its activity. However, the contribution of each of these proteins in carcinogenesis is still poorly understood.

Colorectal cancer (CRC) is the third most common malignancy worldwide and one of the deadliest cancers. CRC involves a multistep process that includes 3 distinct phases: initiation, progression, and metastasis. Researchers recently showed that MCU expression is increased and associated with poor prognosis in patients with CRC. Up-regulation of MCU increases mito$Ca^{2+}$ uptake to promote mitochondrial biogenesis and to facilitate CRC cell growth in vitro and in vivo [11].

The role of MICU2 in the regulation of MCU activity depends on its binding to $Ca^{2+}$ and is mediated by its binding to MICU [12]. MICU2 is a paralog of MICU1—it shares 42% amino acid identity with MICU1—and contains an amino-terminal mitochondrial targeting sequence and a cytosolic C terminus with 2 EF hands [13]. Interestingly, the expression of MICU2 chiefly depends on MICU1, but the reciprocity has not been proved [14,15]. So far, the contribution of MICU2 to the regulation of mito$Ca^{2+}$ signaling, bioenergetics, metabolic reprogramming, mitochondrial dynamics, and CRC development has not been addressed.

Here, we observed that MICU2 expression and the MICU2/MICU1 ratio in patients with CRC are tightly correlated to CRC aggressiveness and stage. For the first time, we demonstrated that MICU2 regulates proliferation and migration of CRC cells in vitro and in vivo. We demonstrated the role of MICU2 on mito$Ca^{2+}$ uptake and its contribution as a crucial element of mitochondrial function, regulating fusion-fission processes, ensuring the proper use of pyruvate by mitochondria and oxidation of fatty acid, and allowing the proper functioning of respiratory chain complexes. Overall, we report for the first time that MICU2 is a guardian of mitochondrial respiratory chain function and a pivotal element during the cancer metabolic switch from oxidative metabolism (oxidative phosphorylation [OXPHOS]) to glycolysis. Moreover, our data suggest that the fine regulation of the MICU1–MICU1 and MICU2–MICU1 dimers in association with MCU may shape the major metabolic transformations in cancer.

## Results

### MICU2 expression is higher in stage IV CRC and metastases

A recent investigation demonstrated that the MCU complex can be linked to overall aggressiveness of CRC [16]. In particular, using The Cancer Genome Atlas Colon Adenocarcinoma

(TCGA-COAD) data set, the authors observed that the expression of MICU1 and MICU2 are respectively down- and up-regulated with stages. The stoichiometry of regulatory units of MCU complex, MICU1 and MICU2, have been shown to modulate the uniporter-mediated $Ca^{2+}$ influx [12].

Here, we focused on investigating the expression of MICU1, MICU2, and the MICU2/ MICU1 ratio in colon tumors using the cohort published by TCGA. As described previously, we found that MICU2 expression is significantly up-regulated in colon tumors of patients with stage IV CRC. MICU3 is not expressed in the CRC. Interestingly, the MICU2/MICU1 ratio is significantly up-regulated in colon tumors of stage IV compared with tumors belonging to less aggressive stages (S1A Fig). In another cohort (GSE41252) encompassing primary tumors but also liver and lung metastatic CRC samples, we observed that the expression of MICU1 is down-regulated while both MICU2 expression and the MICU2/MICU1 ratio are significantly up-regulated in liver metastasis of CRC (Figs 1B and S1B). In addition, the expression of MCU complex players according to CRC stages was verified on TCGA data. Significantly, MCU, MICU1 and MCUb were expressed more weakly compared to healthy tissue, while MICU2 was more expressed (Fig 1B). To further our research, we analyzed proteomic data from 93 patients [17] from the TCGA cohort. Proteomic analysis revealed that among the MICU1, MICU2, EMRE, and MCU components, only MICU2 tended to be overexpressed in stage 4 patients compared to those in stages 1 to 3 ($P = 0.063$). Furthermore, MICU2 is the only component significantly associated with vascular invasion and showed a trend towards association with the presence of distant tumor metastases ($P = 0.077$). Reinforcing this idea, MICU2 is the only component whose elevated expression is significantly correlated with poorer patient survival ($P = 0,039$) (S1E Fig).

Thus, the MICU2/MICU1 ratio is positively correlated to the aggressiveness of a CRC tumor, suggesting that the stoichiometry of the regulatory units of MCU complex may be critical for the tumor development.

Using western blotting, we compared the protein expression of MCU, MICU1, and MICU2 in a non-tumor colorectal epithelial cell line, NCM356, and in 5 CRC cell lines (HT29, DLD1, SW480, HCT116, and SW620) with different mutation and aggressiveness profiles. The tumor cell lines showed significantly higher expression of MCU and MICU1 than the non-tumor cell line. Interestingly, MICU2 expression was significantly higher in the more aggressive tumor cell lines (SW480, HCT116, and SW620) compared with the less aggressive cell lines (HT29 and DLD1) and the non-tumor cell line (NCM356) (Fig 1C). As MICU2 cannot bind to the MCU complex without MICU1, we modified the HCT116 cell line with CRISPR-Cas9 technology to generate 2 MICU2 knockout (KO) clones (Figs 1D and S1C) to modulate the MICU2/MICU1 ratio. Deletion of MICU2 did not alter the protein expression of MCU and MICU1 (Fig 1D), but MICU1 messenger RNA (mRNA) expression was significantly reduced by 25% in one of the MICU2 KO clones (S1C Fig). As MICU2 is a major protein in the uniporter MCU, we tested the consequences of MICU2 loss on these integral membrane subunits of the uniporter complex using native blue polyacrylamide gel electrophoresis/immunoblotting (BN-PAGE). We validated our uniporter specificity by comparing HCT116 WT with HCT116 MCU KO. We then compared HCT116 WT to HCT116 MICU2 KO. We noted no significant changes in the band around 700 kDa but observed a significant decrease in the 450 kDa complex in MICU2 KO cells, accompanied by a marked increase in a 300 kDa protein complex (Fig 1E). These results suggest that the loss of MICU2 leads to structural alterations in the MCU complex.

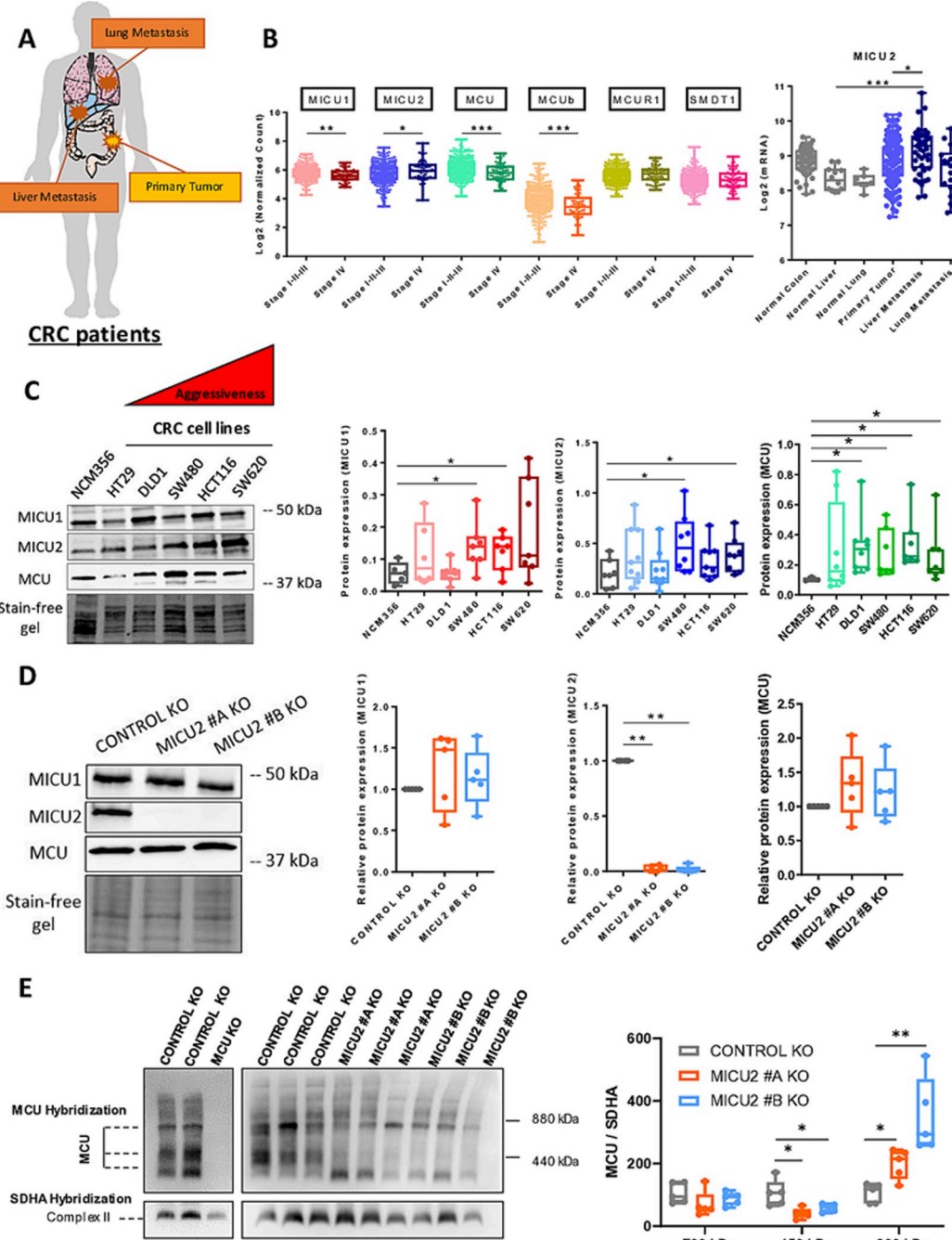

**Fig 1. Expression of MICU2 in stage IV CRC and in metastases from patients with CRC.** (A) Representative scheme of organs used for the bioinformatics analysis of normal, primary tumor, and metastatic samples. (B) Transcriptomic analysis of MICU2 in the GSE41258 (top panel) data sets and of the expression of the players in the MCU complex in the TCGA (bottom panel). Each data point represents an individual sample (ANOVA followed by Dunn's multiple comparisons test). (C) Left panel: representative western blot of MCU, MICU1, and MICU2 in a panel of CRC cell lines with variable aggressiveness. Right panel: quantification of the protein expression of MCU, MICU1, MICU2 western blot ($n$ = 5–9, Mann–Whitney test). (D) Representative western blot of MCU, MICU1, and MICU2 expression in the HCT116 Control and MICU2 KO cell lines obtained by CRISPER-Cas9 methodology. Quantification of the protein expression of MICU1, MICU2, and MCU in the HCT116 Control KO, MICU2 #A KO, and MICU2 #B KO cell lines ($n$ = 6, Kruskal–Wallis test). On all plots, $*p < 0.05$, $**p < 0.01$, and $***p < 0.001$. (E) Left panel: representative western blot of the MCU complex, in HCT116 control, MCU KO and MICU2 KO cell lines. Right panel: quantification of MCU multiprotein complex size by western blot ($n$ = 5, Mann–Whitney test). The data underlying the graphs shown in the figure can be found in S1 and S2 Datas. CRC, colorectal cancer; KO, knockout; MCU, mitochondrial calcium uniporter.

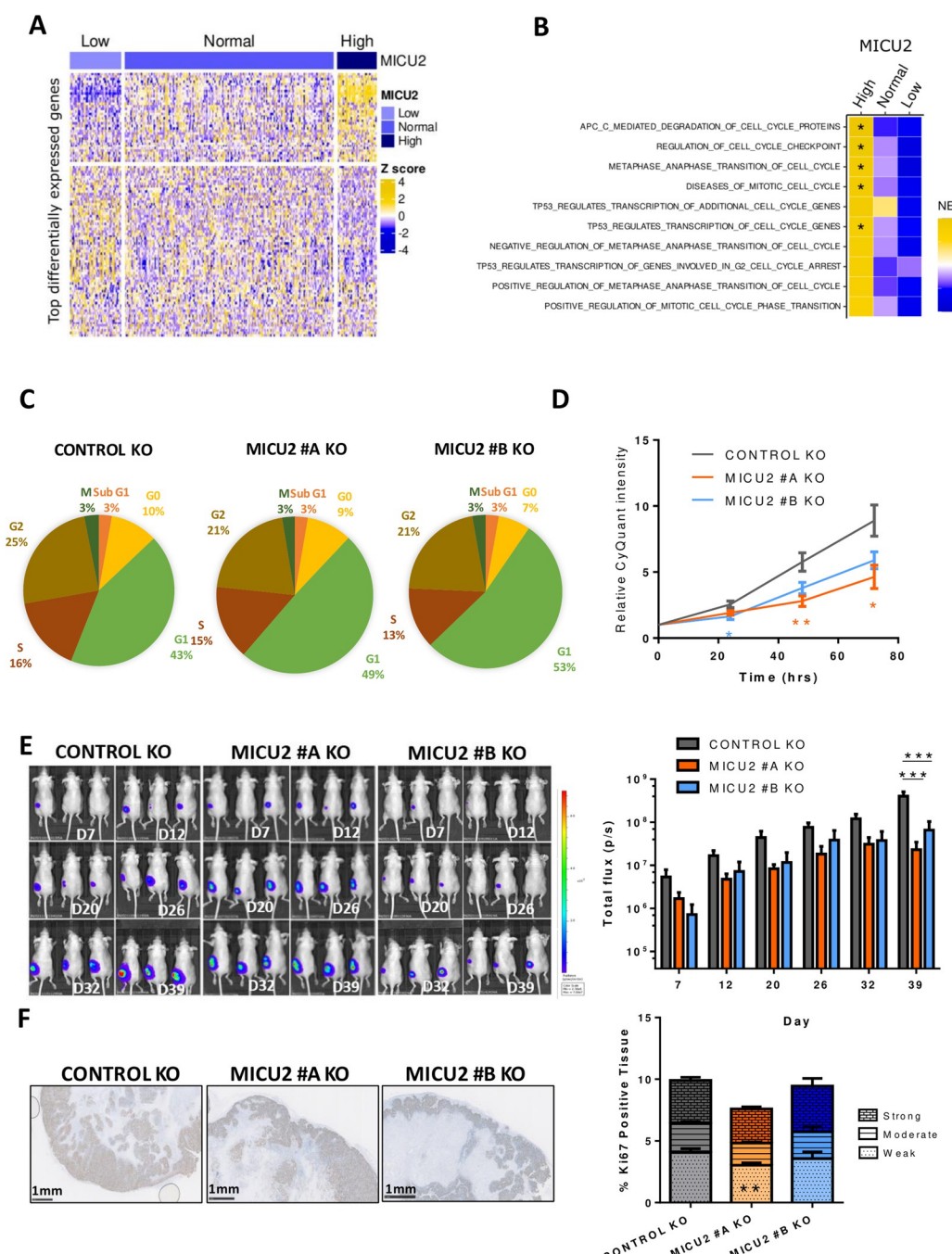

**Fig 2. Involvement of MICU2 in the CRC tumor phenotype.** (A) Heatmap of the most differentially expressed genes between colon tumor samples in the TCGA-COAD dataset with low, normal, or high expression of MICU2. (B) Heatmap of normalized enrichment scores of the top 10 cell cycle-associated gene signatures obtained by GSEA for primary colon tumor samples with low, normal, or high expression of MICU2. (C) Graphical representation of the cell cycle in the Control and MICU2 KO cells lines ($n$ = 6, ANOVA followed by Dunnett's multiple comparisons test). (D) CyQuant proliferation analysis of the Control and MICU2 KO cells lines ($n$ = 11, Kruskal–Wallis test). (E) Representative bioluminescence images of cancer progression in mice injected with the luciferase-expressing Control or MICU2 KO cells lines and quantification of whole-body bioluminescence ($n$ = 10, ANOVA followed by Dunnett's multiple comparisons test). (F) Representative images of tumoral tissue sections stained for Ki67 from xenografted Control and MICU2 KO cells lines and quantification of the percentage of tissue positive for Ki67. The intensity of positivity was scored as weak, moderate, or strong. The scale bar is 1 mm ($n$ = 6, ANOVA followed by Sidak's multiple comparisons test). On all plots, $^*p < 0.05$, $^{**}p < 0.01$, and $^{***}p < 0.001$. The data underlying the graphs shown in the figure can be found in S1, S2 and S3 Datas. CRC, colorectal cancer; GSEA, gene set enrichment analysis; KO, knockout.

## MICU2 regulates CRC tumor growth and metastasis formation in vitro and in vivo

Primary tumors of TCGA-COAD were classified as "High" or "Low" for MICU1, MICU2, and the ratio of MICU2/MICU1 if the values of their expression/ratio are respectively superior to the 95th percentile or inferior to the 5th percentile of the expression/ratio in normal samples (S1D Fig). Using this classification of tumors samples, we then performed a differential expression analysis (DEA) followed by a gene set enrichment analysis (GSEA) to identify genes and pathways that are differentially modulated based on the status of MICU2 or the MICU2/MICU1 ratio (Figs 2A and S2A). Interestingly, we observed that cell cycle-related gene sets are frequently enriched in tumor samples with high MICU2 expression and tend to be less represented in tumor samples with a low MICU2 expression (Figs 2B and S2B).

Cell cycle analysis by flow cytometry demonstrated an increase in the percentage of cells in the G1 phase associated with a decrease in proliferation of the MICU2 KO cell lines compared with the Control KO cell line (Fig 2C and 2D). The HCT116 control and MICU2 KO cell lines were engineered to endogenously express luciferase and injected subcutaneously into NOD-SCID mice. There was a significant reduction in total bioluminescence in mice injected with HCT116 MICU2 KO cells compared with mice injected with the control HCT116 cells (Fig 2E). To confirm the role of MICU2 in CRC tumor growth, we performed Ki67 staining on tumor sections. We noted a significant reduction in cell proliferation for MICU2 #A KO tumors but not for MICU2 #B KO tumors (Fig 2F). This outcome could be potentially explained by the delay we observed for the tumor engraftment of MICU2 #B KO cells. Hence, at the end of our experiment, these tumors were still in the exponential growth phase.

To further investigate the role of MICU2 in cancer, we investigated its role in the response to CRC chemotherapy treatment. We evaluated the dose-response of oxaliplatin and 5-fluorouracil (5FU) in the Control and MICU2 KO cell lines. Both cell lines showed a concentration-dependent reduction in cell viability, but without a significant difference in the sensitivity to these chemotherapeutic agents (S2C Fig). Based on the effect of MCU that has been described in cancer cells, we evaluated the role of MICU2 in cell motility (wound healing and spheroid migration assays). We noted decreased cell migration in the MICU2 KO cell lines compared with the Control KO cell line (S2D and S2E Fig). CRC metastatic dissemination to the liver is one of the most life-threatening malignancies in humans and represents the leading cause of CRC-related mortality. Forced metastasis models are performed by injecting HCT116 WT or MICU2 KO cells into the spleen. To prevent primary tumor growth in the spleen, a hemi-splenectomy is performed 15 min after injection. Using an in vivo imaging system, we observed a significant reduction in total bioluminescence in nude mice injected with HCT116 MICU2 KO cells compared to mice injected with HCT116 WT cells (S2F and S2G Fig). Additionally, the number of metastatic nodules in liver was higher in mice injected with HCT116 WT cells than in those injected with HCT116 MICU2 KO cells (S2H Fig).

Taken together, these results demonstrate that MICU2 expression is associated with several cancer cell properties. In particular, MICU2 is critical for cancer cell proliferation and the cell cycle. MICU2 is crucial for the migration of colon cancer cells and metastatic spread to the liver.

## MICU2 protects CRC cells against fragmentation and controls mitoCa$^{2+}$ uptake

Mitochondria share different contact sites with the endoplasmic reticulum (ER) and the plasma membrane [18,19]. During the movement of $Ca^{2+}$ from the extracellular or intracellular compartments, the mitoCa$^{2+}$ uptake machinery shapes the cellular response. The mutual

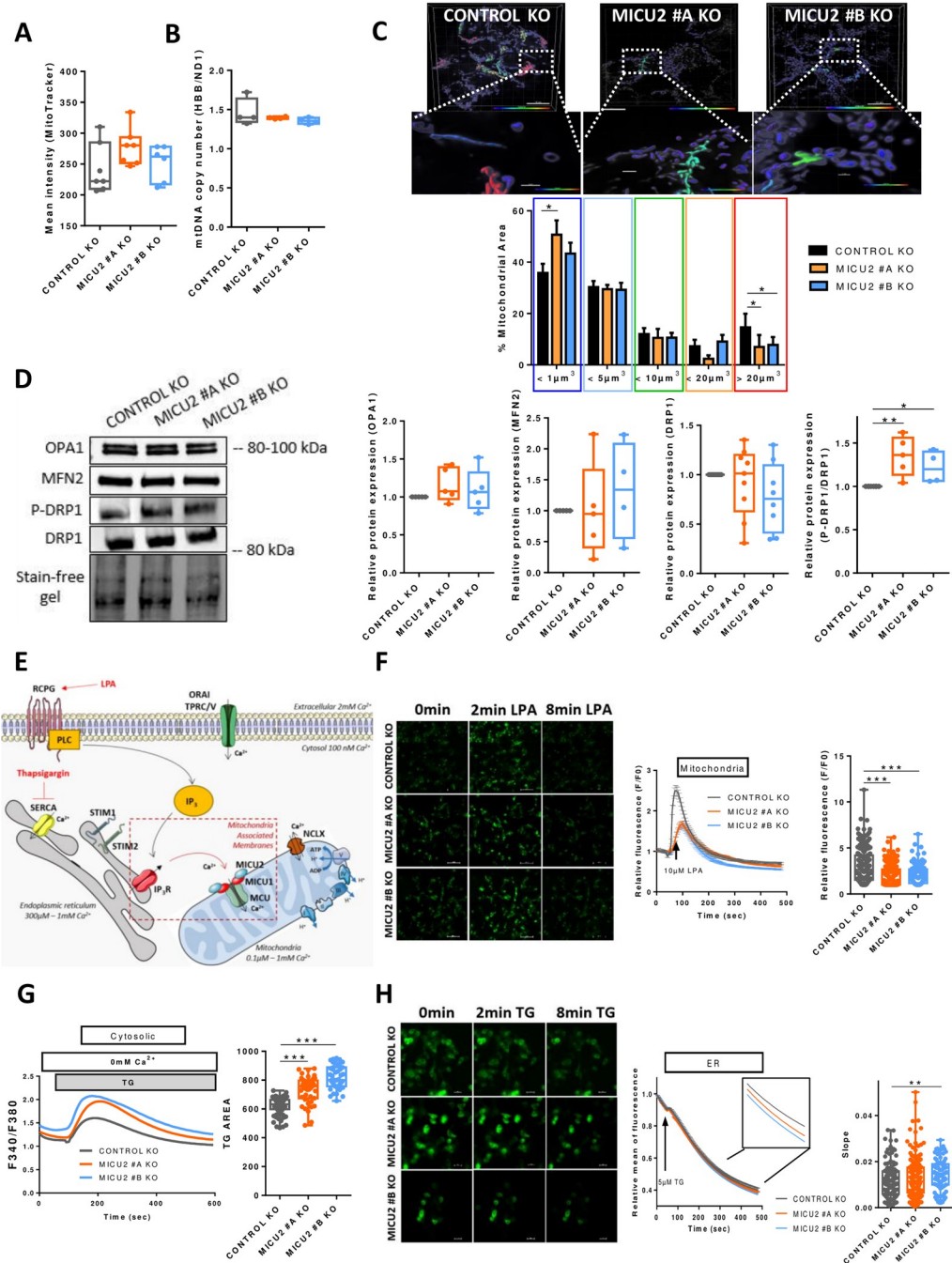

**Fig 3. The role of MICU2 in the mitochondrial network and Ca²⁺ signaling.** (A) Quantification of the number of mitochondria in the Control and MICU2 KO cell lines with Mitotracker green. (B) Quantification of mitochondrial DNA copy number by RT-qPCR using primers HBB and ND1. (C) Top panel: representative 3D images of the mitochondrial network of the HCT116 Control and MICU2 KO cell lines colored based on the mitochondrial volume. The enlargement shows an example of mitochondria with different shapes. The scale bar is 10 μm. Bottom panel: bar graphs representing the distribution of the mitochondrial population based on the mitochondrial volume. The data are presented as the mean ± standard deviation. ANOVA followed by Dunnett's multiple comparisons test with 10 to 15 cells. (D) Left panel: representative western blot of mitochondrial fusion (OPA1 and MFN2) and fission (DRP1 and P-DRP1) proteins in the Control and MICU2 KO cell lines. Right panel: relative protein expression of mitochondrial fusion and fission proteins ($n$ = 4–5, ANOVA followed by Dunn's multiple comparisons test). (E) Schematic illustration of LPA-induced or TG-induced cytosolic Ca²⁺ and mitoCa²⁺ signaling in non-excitable cells. (F) Left panel: representative images of the Control and MICU2 KO cell lines transfected with the genetically encoded mitoCa²⁺ indicator mt-riG6m and stimulated with LPA. The scale bar is 100 μm. Right panel: relative Ca²⁺ responses induced by LPA in Control and MICU2 KO cell lines expressing the mitoCa²⁺ probe mt-riG6m (the data are presented as the mean ± SEM of 8 independent experiments regrouping 354–594 cells, Kruskal–Wallis test). (G) Left panel:

representative SOCE traces in the Control and MICU2 KO cell lines. Right panel: boxplot representing the ER-$Ca^{2+}$ store depletion by TG in the absence of extracellular $Ca^{2+}$ ($n = 6$, $N = 42–95$, ANOVA followed by Dunnett's multiple comparisons test). (H) Left panel: representative images of the Control and MICU2 KO cell lines transfected with the genetically encoded ER $Ca^{2+}$ indicator miGer and stimulated with TG. The scale bar is 20 μm. Right panel: relative ER $Ca^{2+}$ content. The boxplot represents the slope of TG-induced ER $Ca^{2+}$ release ($n = 6–8$, $N = 88–156$, Mann–Whitney test). On all plots, $*p < 0.05$, $**p < 0.01$, and $***p < 0.001$. The data underlying the graphs shown in the figure can be found in S1 Data. ER, endoplasmic reticulum; KO, knockout; LPA, lysophosphatidic acid.

interplay between $Ca^{2+}$ homeostasis and mitochondrial morphology appears to be an essential element in mitochondrial dynamics and $Ca^{2+}$ signaling [20].

To understand the impact of MICU2 KO on the mitochondrial morphology and network, we first evaluated the number of mitochondria and the amount of mitochondrial DNA in the Control and MICU2 KO cell lines. There were no differences in these 2 parameters between the cell lines (Fig 3A and 3B). We then assessed the network and size of mitochondria using confocal imaging. Interestingly, the MICU2 KO #A line had a significantly increased number of mitochondria with a surface area of <1 μm3. Smaller MICU2 KO cell lines have a significantly reduced number of mitochondria with a surface area >20 μm3 compared to the Control KO cell line. This suggests that there is a decrease in the mitochondrial connection network and therefore greater fragmentation (Fig 3C). To confirm this hypothesis, we analyzed proteins linked to mitochondrial dynamics, namely the expression of OPA1 and MFN2, which are involved in fusion, and the expression of DRP1 and its active phosphorylated form (P-DRP1 Ser616), which is involved in fission. There was no difference in the expression of OPA1 or MFN2, but the P-DRP1/DRP ratio was increased in MICU2 KO cell lines, indicating increased mitochondrial fission, consistent with fragmentation of the mitochondrial network (Fig 3D). The mRNA expression of fission (DRP1 and FIS1) and fusion (OPA1 and MFN2) genes was not significantly altered in MICU2 KO cell lines compared to the Control KO cell line (S3A Fig).

*MICU2* modulates mito$Ca^{2+}$ uptake. Thus, we examined the consequences of the genetic deletion of MICU2 on mitochondrial, cytosolic, and ER $Ca^{2+}$ signaling. Stimulation of a G protein–coupled receptor (GPCR) such as the lysophosphatidic acid (LPA) receptor by LPA induces inositol trisphosphate ($IP_3$) production. $IP_3$ then activate $IP_3$ receptors leading to a $Ca^{2+}$ release from ER and induce mitochondria $Ca^{2+}$ influx through the uniporter MCU (Fig 3E).

We assessed mito$Ca^{2+}$ homeostasis by using a genetically encoded $Ca^{2+}$ indicator targeting the mitochondrial compartment (mt-riG6m). Stimulation of cells with LPA induced a transient increase in mito$Ca^{2+}$ that was significantly reduced in the MICU2 KO cell lines compared with the Control KO cell line (Fig 3F). We then wondered whether this reduction in mito$Ca^{2+}$ influx might affect cytosolic $Ca^{2+}$ homeostasis. Stimulation of transitory ER $Ca^{2+}$ release by LPA activates store-operated calcium entry (SOCE) [21]. MICU2 KO had no effect on SOCE induced by LPA (S3B Fig). Interestingly, thapsigargin (TG)-mediated release of ER $Ca^{2+}$ stores were increased in the MICU2 KO cell lines compared with the Control KO cell line without modifying SOCE (Figs 3G and S3C). We then assessed whether this change is due to higher $Ca^{2+}$ content in ER stores. Using the genetically encoded $Ca^{2+}$ indicator targeting ER (miGer), we observed a small but significant increase in the speed of the TG-induced ER $Ca^{2+}$ release in the MICU2 KO #B cell line (Fig 3H). To investigate whether the presence of MICU2 could alter the buffering capacity of cytosolic calcium, potentially affecting functions such as cell proliferation or migration, we studied the impact of cytosolic calcium reduction using an intracellular $Ca^{2+}$ chelator (BAPTA-AM). Our results demonstrate that the presence of BAPTA-AM reduces the viability of both WT and MICU2 KO colonic cells (S3D Fig). This

suggests that the presence or absence of MICU2 does not play a significant role in controlling cytosolic calcium concentration during proliferative activity.

Crosstalk between $Ca^{2+}$ and reactive oxygen species (ROS) is found in many pathologies including cancers [22]. Hence, we measured ROS production in HCT116 Control and MICU2 KO clones. We analyzed the levels of total superoxide ions (DHE) and mitochondrial superoxide (MitoSox) and the amount of hydrogen peroxide (DCFDA). We observed no difference between the Control and MICU2 KO cell lines (S3E Fig). In addition, there was no difference in the viability of the Control and MICU2 KO cell lines treated with a mitochondria-targeted antioxidant (MITOTEMPO) (S3F Fig). This outcome confirms that genetic deletion of MICU2 does not affect ROS production.

Taken together, these results reveal that MICU2 is necessary for the stability and the quality of mitochondrial network while allowing the proper management of ER-mitochondria $Ca^{2+}$ flux.

## Deletion of MICU2 reduces respiratory complex expression and activity associated with loss of respiratory chain sensitivity to $Ca^{2+}$

*MitoCa*$^{2+}$ plays critical signaling roles in regulating ATP production linked to oxygen consumption by mitochondria. We hypothesized that the deletion of MICU2 alters the functioning of the respiratory chain.

First, we tested this hypothesis by analyzing in TCGA-COAD data set based on the expression of MICU1, MICU2, and the MICU2/MICU1 ratio. The mRNA expression of genes belonging to the mitochondrial respiratory chain complexes are not modified by the expression of MICU1, MICU2, or the MICU2/MICU1 ratio (S4A Fig). However, we observed that TCGA-COAD tumor samples with a "high" MICU1 status show higher expression of genes associated with the assembly of complexes I and IV of the mitochondrial respiratory chain compared with tumor samples with a "low" MICU1 status (Figs 4B and S4B). In contrast, we observed that TCGA-COAD tumor samples with a "high" status of MICU2 and the MICU2/MICU1 ratio present higher expression of genes associated with the assembly of complexes of the mitochondrial respiratory chain compared with tumor samples with a "low" status for MICU2 and the MICU2/MICU1 ratio (Figs 4B and S4B), suggesting that the expression of MICU2 may contribute to the assembly complexes or supercomplexes. To validate this finding, we analyzed their expression and assembly into supercomplexes by Blue-Native PAGE. We observed a reduction in assembly of the $I_1III_2IV$ supercomplex related to a significant reduction in the quantity of complex IV (Fig 4C). Consistently, the maximal activity of complex IV (cytochrome *c* oxidase) in the MICU2 KO cell lines was decreased compared with the Control KO cell line (Fig 4D). Mitochondria is the main organelle supporting energy supply of cells [23]. The reduced expression of the terminal enzyme of the respiratory chain, complex IV, in MICU2 KO cells could limit the mitochondrial oxidative capacity. Inhibition of complex I and ATP synthase with rotenone and oligomycin, respectively, had a more pronounced effect on the viability of the Control KO cell line than the MICU2 KO cell lines (Fig 4E), suggesting that the MICU2 KO cell line is less dependent on oxidative phosphorylation (OXPHOS). In parallel, we verified that oligomycin and rotenone treatment did not significantly modify the mRNA expression of MICU1, MICU2, or MCU (S4C Fig).

Variation in the mitoCa$^{2+}$ concentration can have a dual role on OXPHOS activity. First, it can directly modify the activity of respiratory complexes. Second, $Ca^{2+}$ can activate the tricarboxylic acid (TCA) cycle by affecting isocitrate dehydrogenase and α-ketoglutarate dehydrogenase, thus increasing OXPHOS. Using permeabilized cells, we evaluated the consequences of different cytosolic $[Ca^{2+}]$—0, 150 nM, 1 μm, and 8.4 μm—on OXPHOS stimulated by succinate, to avoid the consequence of $Ca^{2+}$ variation on the activity of TCA cycle and focus only

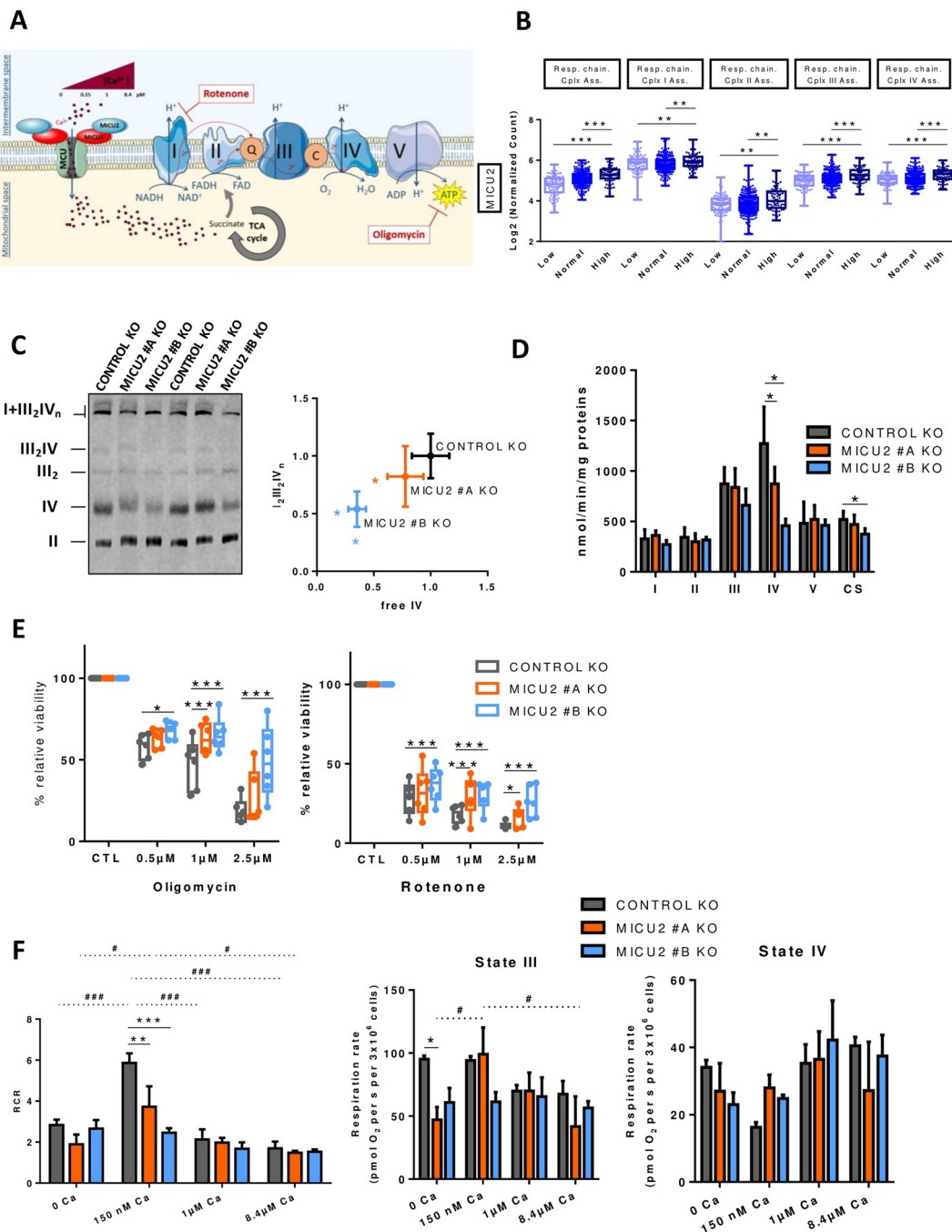

**Fig 4. Contribution of MICU2 to respiratory chain complexes.** (A) Simplified diagram of the respiratory chain and ATP synthesis in the mitochondria by OXPHOS. The production of ATP is possible due to the formation of a proton gradient around the membrane generated by the energy of the electrons supplied by NADH and $FADH_2$ to this chain. These electrons are transported via different complexes: I, II, III, and IV. This diagram also illustrates the analysis of mitochondrial cellular respiration of different $[Ca^{2+}]$ (0, 150 nM, 1 µm, and 8.4 µm). $Ca^{2+}$ can regulate succinate, which in turn regulates complex II. (B) Boxplot representing the average expression of genes associated with the assembly of mitochondrial respiratory chain complexes I, II, III, and IV based on the status of MICU2 in primary colon tumor samples from the TCGA-COAD data set. (C) Analysis of supercomplex and complex assembly through BN-PAGE followed by western blotting. Left panel: representative blot hybridized with anti-NDUFS2 (complex I), UQCRC2 (complex III), SDHA (complex II), and MT-CO1 (complex IV). The positions of the different free complexes or supercomplexes are indicated. Right panel: the supercomplex I1III2IVn quantity normalized to the CII quantity (y-axis) according to the free CIV quantity, normalized to the CII quantity (x-axis). The data are presented as the mean ± standard deviation ($n$ = 6, Mann–Whitney test). (D) Maximal activities of the respiratory chain complexes I (NADH ubiquinone reductase), II (succinate ubiquinone reductase), III (ubiquinol

cytochrome *c* reductase), and complex IV (cytochrome oxidase), and activity of complex V (F1-ATPase) and the TCA enzyme CS. The data are presented as the mean ± standard deviation ($n = 5$, Mann–Whitney test). (E) Boxplots representing the viability of the Control and MICU2 KO cell lines cultured in the presence of oligomycin (complex V inhibitor) (left panel) or rotenone (complex I inhibitor) (right panel) at increasing doses (0.5, 1, and 2.5 μm) for 48 h ($n = 6$, ANOVA followed by Dunnett's multiple comparisons test). (F) Barplots representing respiratory states III, IV and RCR defined by the difference in oxygen consumption between succinate and substrate-free respiration (state II), phosphorylating respiration measured after injection of ADP and succinate (state III), and oxygen consumption after injection of oligomycin (state IV). The RCR is the respiratory control ratio calculated by the ratio between state III and IV. Respiratory rates are plotted against different [Ca2+] levels (0, 150 nM, 1 μm, and 8.4 μm) ($n = 6$–$9$, ANOVA followed by Dunnett's multiple comparisons test). On all graphs, $^*p < 0.05$, $^{**}p < 0.01$, and $^{***}p < 0.001$. The data underlying the graphs shown in the figure can be found in S1 and S2 Datas. BN-PAGE, blue native polyacrylamide gel electrophoresis; CS, citrate synthase; TCA, tricarboxylic acid.

on the role of $Ca^{2+}$ on the respiratory chain [24]. We were particularly interested in state III of mitochondrial respiration stimulated by adenosine diphosphate (ADP) and state IV which is the inhibition of ATP synthase mainly controlled by proton leakage in the presence of substrate. So we defined the respiratory chain ratio (RCR) as an indicator of proper functioning of the respiratory chain [25].

Globally, no significant effect of either the variation of the cytosolic $[Ca^{2+}]$ or the deletion of MICU2 is observed on both state III or state IV. However, the optimal function of the respiratory control ratio (RCR) is observed at physiological cytosolic $[Ca^{2+}]$ (150 nM). At this concentration, the deletion of MICU2 significantly impairs the RCR. In absence of cytosolic $[Ca^{2+}]$ or at supra-physiological cytosolic $[Ca^{2+}]$, we observe a reduced RCR and no significant effect of the deletion of MICU2 on the RCR. MICU2 optimizes mitochondrial respiration under physiological $[Ca^{2+}]$.

Overall, we demonstrated that MICU2 controls the complex biogenesis and activity of the respiratory chain and its presence optimizes oxygen consumption linked to mitochondrial ATP production in an environment with a physiological cytosolic $[Ca^{2+}]$. We therefore took a closer look at metabolic rewiring and the substrate supply pathways regulated by calcium.

## MICU2 controls proliferation of CRC cells by promoting fatty acid oxidation and mitochondrial pyruvate utilization

Therefore, we studied the cellular respiration of intact Control and MICU2 KO cells, in standard RPMI medium with the presence of glucose (2 g/l) and glutamine (0.4 g/l). We observed a significant reduction in the basal respiration rate (oxidative metabolism), the respiration dedicated to ATP production-linked oxygen consumption, and the maximal respiration (maximal oxidative capacity) in the MICU2 KO cell lines compared with the Control KO cell line (Fig 5A), which could be linked to complex IV deficiency. However, mitoCa$^{2+}$ allows adjusting ATP production-linked oxygen consumption to the cellular energy demand by directly or indirectly regulating PDH, isocitrate dehydrogenase, and α-ketoglutarate dehydrogenase activities. Thus, the decreased oxidative metabolism and mitochondrial ATP synthesis could also be due to limited substrate supply. We addressed the contribution of MICU2 to the preferred pathways and substrates (glucose, glutamine, and fatty acids) used for energy production by the mitochondria. Then, we evaluated the importance of the 3 metabolic pathways separately.

Fatty acids are transported into the cytosol and mitochondria by CD36 and carnitine palmitoyltransferase 1 (CPT1)/CPT2, respectively. Once inside the mitochondria, fatty acids are degraded by beta-oxidation to produce, among other things, acetyl-CoA that fuels the TCA cycle and energy production (Fig 5B). Here, using Oil Red O staining, we observed an increase in lipid droplets in the MICU2 KO cell lines compared with the Control KO cell line, suggesting either greater fatty acid storage or reduced beta-oxidation (Fig 5C). There was no difference between the Control and MICU2 KO cell lines in the mRNA expression of perilipins

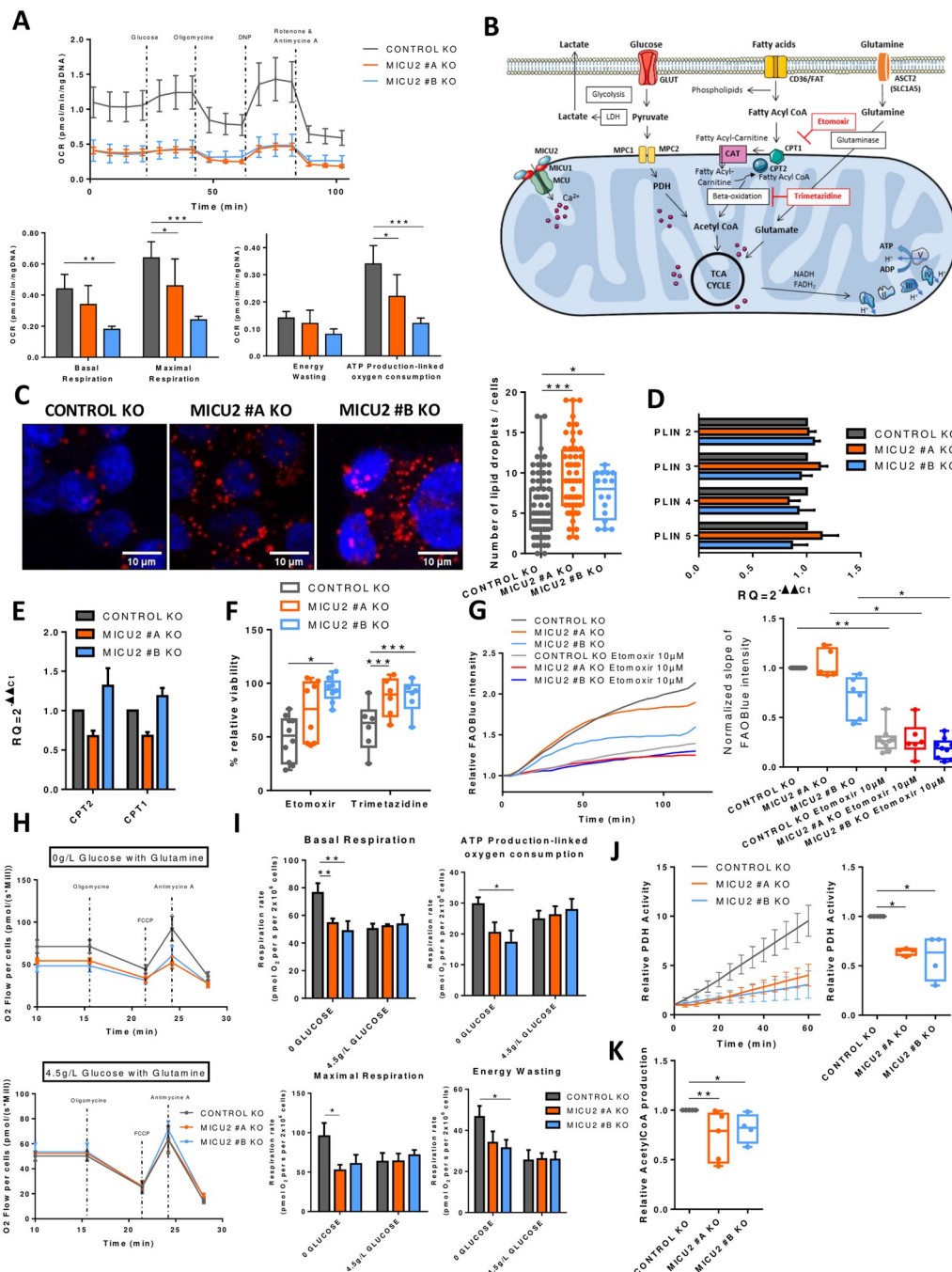

**Fig 5. MICU2 modulates substrate selection for the TCA cycle.** (A) Top panel: graph representing the OCR as a function of time in the Control and MICU2 KO cell lines and with sequential addition of glucose, oligomycin, DNP (protonophore), antimycin A (complex III inhibitor), and rotenone. Bottom panel: bar plots representing the basal respiration, maximal respiration, proton leak, and ATP production in Control and MICU2 KO cell lines (*n* = 5, ANOVA followed Sidak's multiple comparisons test). (B) Illustration of the different metabolic pathways essential for proper functioning of the mitochondria through the TCA cycle. (C) Left panel: representative images of lipid droplets (red) and nuclei (blue) in the Control and MICU2 KO cell lines. The scale bar is 10 μm. Right panel: boxplot representing the number of lipid droplets per cell (*n* = 17–80 cells, Mann–Whitney test). (D) qRT-PCR data showing fold changes in mRNA levels of fatty acid degradation proteins in the Control and MICU2 KO cell lines (*n* = 8). (E) qRT-PCR data showing fold changes in mRNA levels of carnitine palmitoyltransferase 1 and 2 (CPT1 and CPT2) proteins in the Control and MICU2 KO cell lines. The results are expressed as the mean ± SEM of 4 and 5 independent experiments. (F) Boxplot representing the viability of the Control and MICU2 KO cell lines cultured for 48 h in the presence of 10 μm etomoxir (CPT1a inhibitor) or 10 μm trimetazidine (beta-oxidation inhibitor) (*n* = 6–9, ANOVA followed by Dunnett's

multiple comparisons test). (G) Left panel: representative curve of FAOBlue fluorescence relative intensity in the Control and MICU2 KO cell lines with or without 10 μm etomoxir. Right panel: boxplot-representing slope of FAOBlue fluorescence relative intensity with or without etomoxir (10 μm). The data are presented as the mean ± standard error ($n$ = 5–9, Kruskal–Wallis test). (H) Graph representing the oxygen consumption rates of the Control and MICU2 KO cell lines in the absence with glutamine (top panel) or in the presence of high glucose (4.5 g/l) and glutamine (bottom panel) in basal conditions and with sequential addition of oligomycin, FCCP (uncoupler), and antimycin A (complex III inhibitor). (I) Bar plots representing the basal respiration and ATP production-linked oxygen consumption (top panel) with maximal respiration and energy wasting (bottom panel) in the Control and MICU2 KO cell lines in presence of glutamine with or without high glucose (4.5 g/l). The data are presented as the mean ± standard error of the mean ($n$ = 6, ANOVA followed Dunnett's multiple comparisons test). (J) Left panel: PDH activity at 60 min in the Control and MICU2 KO cell lines. Right panel: boxplot representing the PDH in the Control and MICU2 KO cell lines ($n$ = 4, ANOVA followed by Dunn's multiple comparisons test). (K) Boxplot representing the production of acetyl-CoA in RPMI culture medium in the Control and MICU2 KO cell lines ($n$ = 4–5, Kruskal–Wallis test followed by Dunn's multiple comparisons test). On all plots, *$p < 0.05$, **$p < 0.01$, and ***$p < 0.001$. The data underlying the graphs shown in the figure can be found in S1 Data. KO, knockout; OCR, oxygen consumption rate; TCA, tricarboxylic acid.

(PLIN), proteins that assemble on the surface of lipid droplets, or of CPT1 and CPT2 (Fig 5D and 5E). Compared with the Control KO cell line, the viability of the MICU2 KO cell lines was not reduced as much by the inhibitors of fatty acid CoA transporters CPT1 and CPT2 (etomoxir) or beta-oxidation (trimetazidine). There was no significant difference in the mRNA expression of MCU, MICU1, and MICU2 after etomoxir or trimetazidine treatment (Figs 5F and S5A). This suggests that in the absence of MICU2, the HCT116 cell line tends to be less dependent of fatty acid metabolism.

We then wondered whether the mitochondrial energy production mediated by fatty acid utilization depends on MICU2 expression. Hence, we used a fluorescent indicator of fatty acid oxidation (FAOBlue), which indicated that the MICU2 KO #B cell line had reduced fatty acid oxidation compared with the Control and MICU2 KO #A cell lines (Fig 5G).

Taken together, these results suggest that fatty acid beta-oxidation is not the preferred energy pathway used by MICU2 KO cells and that the deletion of MICU2 could alter the quantity of lipid droplets and beta-oxidation.

Glutamine is transformed into glutamate to fuel the TCA cycle. Glutaminolysis is widely used in HCT116 cells to sustain TCA cycle anaplerosis [26]. To determine whether MICU2 expression regulate the oxidative metabolism sustain through glutamine oxidation, we first analyzed respiration rates on intact cells in the only presence of glutamine (Fig 5H). With glutamine as substrate, basal respiration rate, respiration-linked to ATP synthesis and maximal oxidative capacity were all significantly reduced in MICU2 KO cells. However, we did not note significant difference in the decrease of the viability of Control and MICU2 KO cells induced by inhibitors of the amino acid transporter (V-9302) and glutaminase (telaglenastat) (S5B Fig). Glutamine has a versatile role in cell metabolism, not only participating in TCA cycle anaplerosis but also the biosynthesis of nucleotides, glutathione (GSH), and other nonessential amino acids. Thus, this suggests that MICU2 KO cells still displayed dependency on glutamine metabolism for their survival, despite a reduced mitochondrial oxidative metabolism support by glutamine.

High glucose concentration (4.5 g/l) is known to force cellular metabolism towards "anaerobic" glycolysis. Indeed, the addition of high glucose decreases oxidative metabolism (basal respiration, approximately 80 nmolO2/sec/million cells in glutamine versus approximately 55 nmolO2/sec/million cells in glutamine and high glucose condition, Fig 5H and 5I) in control cells but has no effect on MICU2 KO cells, suggesting that these cells already have high reliance on glycolytic metabolism.

We thus measured PDH activity (known to be regulated by $Ca^{2+}$) and acetyl-CoA production and observed a significant reduction in the MICU2 KO cell lines compared with the

Control KO cell line (Fig 5J and 5K). We observed a significant increase in PDP1 and PDHA1 mRNA expression in the MICU2 KO cell lines (S5C and S5D Fig), suggesting an increase in the expression of pyruvate regulator alongside a decrease in global PDH activity. To take our analysis a step further, we used mRNA expression to study the pentose phosphate pathway. PRPS1 and PRPS2 were increased in MICU2 KO cell lines (S5E Fig). Jing and colleagues [27] showed that PRPS1 up-regulation is more important in promoting tumorigenesis and is a promising diagnostic indicator of CRC.

Overall, our results indicate that the MICU2 KO cell lines have lower OXPHOS associated with a reduction in PDH activity and acetyl-CoA production. Interestingly, the balance between glucose, beta-oxidation and glutaminolysis is modified in MICU2 KO cells.

## MICU2 knockout cells exhibit increased glycolytic flux in vitro and in vivo

Cellular ATP production is dependent of the TCA cycle and glycolysis, both of which are associated with glucose uptake by GLUT transporters. In the cytosol, glycolysis transforms glucose into pyruvate while producing ATP. Then, pyruvate is degraded into lactate by lactate dehydrogenase (LDH) and excreted from the cell by lactate transporters (MCT). Pyruvate can be transported into mitochondria through MPC proteins. PDH produces acetyl-CoA that serves as substrate for TCA cycle and mitochondrial ATP production (Fig 6A).

Here, we observed significantly increased mRNA expression of LDH and lactate transporters (MCT2 and MCT4) in the MICU2 KO cell lines compared with the Control cell line (Fig 6B), suggesting a more pronounced dependence on anaerobic glycolysis in these cell lines. Consistently, the MICU2 KO cell lines showed reduced viability when cultured in presence of a non-metabolizable glucose analog (2-deoxy-D-glucose [2DG]) compared with only a modest effect for the Control KO cell line (Fig 6C). Incorporation of fluorescent glucose (2-NBDG) and the production of lactic acid were also higher in the MICU2 KO cell lines compared with the Control KO cell line (Fig 6D and 6E).

To determine the preference of MICU2 KO cells for a glycolytic phenotype, we treated them with UK5099 (MPC1/MPC2 pyruvate transporter inhibitor). In these conditions, pyruvate cannot be used to sustain oxidative metabolism but is mainly converted to lactate. The Control KO cell line was more sensitive to UK5099 than the MICU2 KO cell lines (Fig 6F). In parallel, we found that UK5099 treatment did not significantly modify the mRNA expression of MICU1, MICU2, or MCU (S6A Fig). This finding suggests that the MICU2 KO cell lines are less dependent on energy generated from pyruvate by mitochondria.

Cancer cells growing in a hypoxic environment tend to be more dependent on anaerobic glycolysis for energy production [28]. As the MICU2 KO cell lines develop a preference for glycolysis, we measured the oxygenation of subcutaneous transplanted MICU2 KO tumors using photoacoustic imaging. We observed a significant reduction in the oxygenation of MICU2 KO #A primary tumors compared with Control KO tumors, suggesting a more hypoxic environment in those tumors (Fig 6G). This was confirmed by the staining of tumor sections with pimonidazole, a hypoxia indicator (Fig 6H). Interestingly, we found a positive correlation between the proliferative index (Ki67) and the pimonidazole only for MICU2 KO #A tumors, suggesting that the growth of MICU2 KO tumors is dependent on hypoxia (Fig 6I).

From the TCGA-COAD data set, we defined 3 gene sets with genes associated with glycolysis, hypoxia, mitochondrial respiration, or the TCA cycle (Figs 6J and S6B). Interestingly, the expression of MICU1 is positively correlated with the average expression of genes associated with glycolysis (S6C Fig). By contrast, MICU2 expression and the MICU2/MICU1 ratio are negatively correlated with gene sets associated with glycolysis and hypoxia and positively correlated with gene sets associated with mitochondrial respiration and the TCA cycle (Fig 6K).

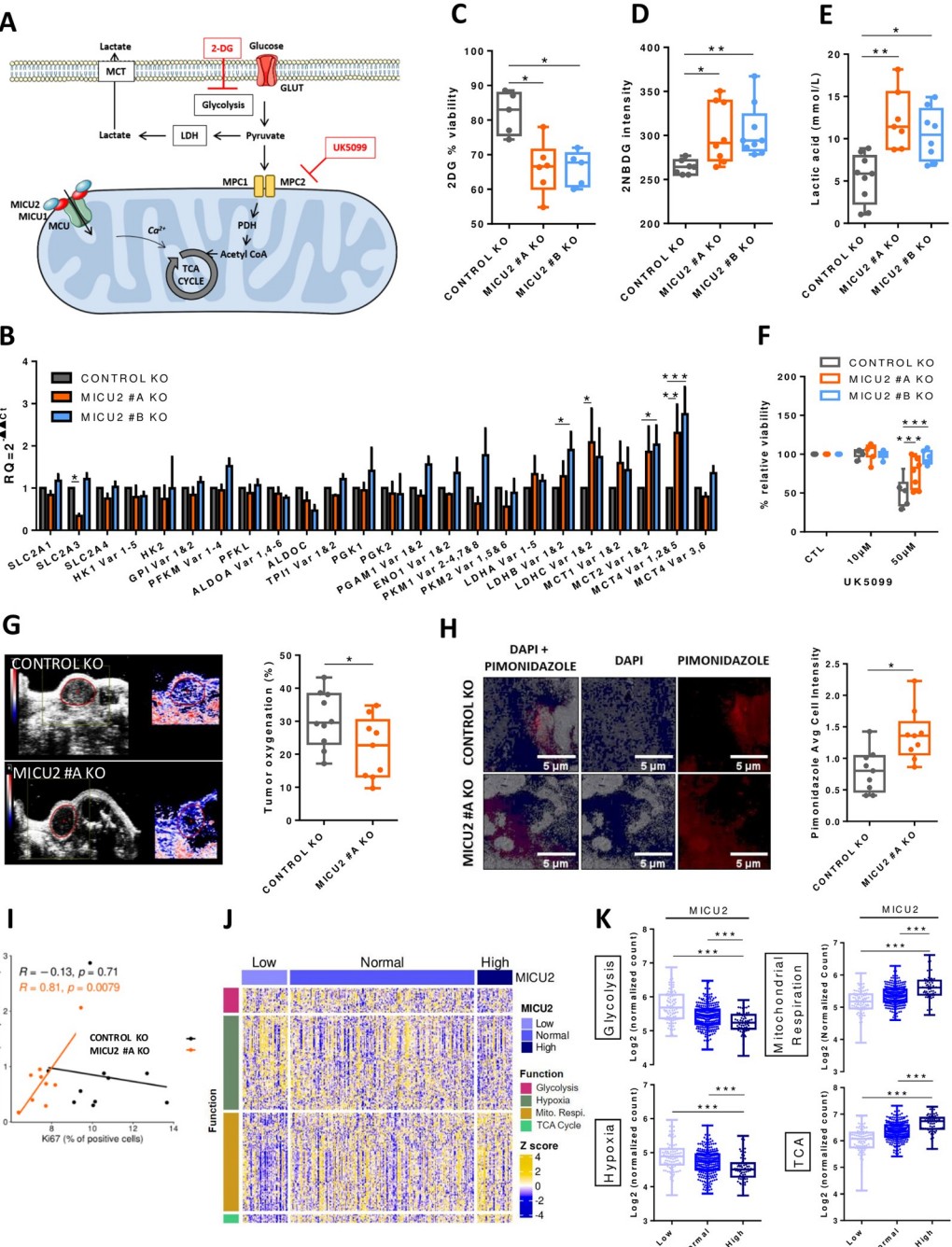

**Fig 6. MICU2 expression correlates negatively with glycolysis and hypoxia and is associated with OXPHOS and the TCA cycle.** (A) Diagram showing the glycolytic pathway. Glucose crosses the plasma membrane via GLUT; pyruvate is then reduced to lactate or can also pass the mitochondrial membrane to produce acetyl-CoA via PDH. Inhibitors of glycolysis such as 2DG (a non-metabolizable glucose analog) and UK5099 are present at their target. (B) Bar plots representing the level of expression of glycolysis-related genes in the Control and MICU2 KO cell lines ($n$ = 3–13, ANOVA followed Dunnett's multiple comparisons test). The data are presented as the mean ± SEM. (C) Boxplot representing viability of the Control and MICU2 KO cell lines after culturing for 48 h with 2DG ($n$ = 5–6, ANOVA followed by Dunn's multiple comparisons test). (D) Boxplot representing the fluorescence intensity of the Control and MICU2 KO cell lines incubated for 30 min with the fluorescent glucose analog 2NBDG ($n$ = 7–9, ANOVA followed by Dunn's multiple comparisons test). (E) Boxplot representing lactic acid production in the Control and MICU2 KO cell lines ($n$ = 7–9, ANOVA followed by Dunn's multiple comparisons test). (F) Boxplot representing viability of the Control and MICU2 KO cell lines cultured in the presence of UK5099 (MPC inhibitor) ($n$ = 6–9, ANOVA followed by Dunnett's multiple comparisons test). (G) Left panel: representative images of photoacoustic imaging in the Control and MICU2

KO cell lines illustrating the oxygen levels in tumors at day 32. Right panel: boxplot representing tumor oxygenation of the Control and MICU2 KO cell lines ($n$ = 9–10, Mann–Whitney test). (H) Left panel: representative images of hypoxic areas in tumors detected by pimonidazole (red) and DAPI (blue) staining. The scale bar is 5 µm. Right panel: boxplot representing the quantification of pimonidazole-positive areas on whole tumor sections ($n$ = 9, Mann–Whitney test). (I) Scatter plot representing the correlation of the Ki67 labeling index versus the pimonidazole score in the Control and MICU2 KO cell lines (R is the Pearson correlation coefficient). (J) Heatmap representing the expression of genes associated with glycolysis, hypoxia, mitochondrial respiration, and the TCA cycle as a function of the MICU2 status of primary tumor samples from the TCGA-COAD data set. (K) Boxplot representing the mean expression of genes associated with glycolysis, hypoxia, mitochondrial respiration, and the TCA cycle according to the MICU2 status of primary tumor samples of the TCGA-COAD data set. On all plots, $^{*}p < 0.05$, $^{**}p < 0.01$, and $^{***}p < 0.001$. The data underlying the graphs shown in the figure can be found in S1 and S2 Datas. KO, knockout; TCA, tricarboxylic acid.

Thus, MICU2 expression seems to promote the expression of genes associated with mitochondrial energy production. Our results suggest that CRC cells with low MICU2 expression develop a preference for glycolysis to support cancer cell proliferation in a hypoxic environment.

## Discussion

In this study, we demonstrated that MICU2 is a guardian of mitochondrial OXPHOS and a pivotal element during the cancer metabolic switch from oxidative metabolism to glycolysis. The MCU complex mediates mitoCa$^{2+}$. Within this complex a scaffold (protein) EMRE and mtCa$^{2+}$ proteins (MICU1, MICU2, and MICU3) interact with MCU, a pore-forming unit able to modulate mitoCa$^{2+}$ absorption. Deregulation of mitoCa$^{2+}$ homeostasis has been described in several cancers. Consistently, MCU expression is elevated in various types of cancer, including breast cancer, hepatocellular carcinoma, melanoma, and CRC [29].

The Ca$^{2+}$-dependent activation of MCU is conferred by MICUs, a set of EF hand–containing regulatory proteins. In all tissues this process is mediated by MICU1 and MICU2: MICU1 is responsible for keeping MCU in a closed state [30] and MICU2 allows Ca$^{2+}$-dependent MCU opening [15]. In brain and muscle, MICU3 replaces MICU2 for this task [31]. Several studies have endeavored to decipher molecular mechanism by which MICU1–MICU2 and MICU1–MICU3 heterodimers or MICU1–MICU1 homodimers regulate MCU channeling. However, the study of MICU proteins in a cancerous context remains very limited. In ovarian cancer, MICU1 silencing inhibits clonal growth, migration, and invasion in vitro, whereas its silencing in vivo inhibits tumor growth and increases cisplatin efficacy and overall survival [32]. Interestingly, inhibition of MICU1 results in the concomitant disappearance of MICU2 protein expression, indicating that MICU2 requires MICU1 for its stability, but not vice versa [15]. Thus, we investigated the stoichiometry of MICU2/MICU1 and the molecular function of MICU2 in cancer cells by knocking down the MICU2 gene in CRC cells. Interestingly, BN-PAGE analysis also reveals that the loss of MICU2 appears to partially alter the structure of the uniporter. This alteration is evident not around the 700 kDa structures but rather between the 300 and 450 kDa structures, where we observe an increased proportion of the 300 kDa form and a decreased amount of the 450 kDa form. Given the absence of differences around 700 kDa and the observed variations between 300 and 450 kDa, it is challenging to determine whether MICU2 loss leads to partial MICU1/MICU1 homodimerization or affects other proteins such as EMRE. Further studies will be necessary to uncover the structural changes in MCU in the absence of MICU2 in our model.

Bioinformatics analysis of publicly available CRC data sets revealed that transcriptomic expression of MICU2 or the MICU2/MICU1 ratio is correlated with the aggressiveness of CRC, with the highest expression measured in stage IV tumors and in metastases of CRC. We have also confirmed through proteomic data analysis that high MICU2 expression

significantly decreases the survival of patients with CRC. These findings confirmed a previous report from Zhu and colleagues [16]. The authors proposed that expression of the different members of MCU complex could be associated with a risk signature that predicts the prognosis and responses to immunotherapy in patients with CRC.

We have provided the first (to the best of our knowledge) experimental evidence that MICU2-mediated mitoCa$^{2+}$ uptake is crucial for CRC cell growth and metastasis formation by stabilizing mitochondrial network (fusion/fission balance), respiratory chain complexes, OXPHOS and regulating main substrate supply pathways in CRC cells. These mechanistic data validate the bioinformatics analysis, which shows that the stoichiometry of the MICU2/MICU1 ratio and in particular the expression of MICU2 are correlated with the expression of genes related to mitochondrial respiration and the TCA cycle and inversely correlated with the expression of genes related to glycolysis in tumor samples.

## MitoCa$^{2+}$ and mitochondria dynamics

We demonstrated that a loss of MICU2 decreases mitoCa$^{2+}$ influx in CRC cells. Recently, Vishnu and colleagues [33] described an equivalent role of MICU2 in mitoCa$^{2+}$ uptake in β cells. The absence of MICU2 has been associated with reduced mitoCa$^{2+}$ uptake, but it also leads to a decrease in the quantity of MCU [13].

In CRC, mitoCa$^{2+}$ uptake promotes TFAM dephosphorylation to enhance mitochondrial biogenesis [34]. Changes in mitochondrial size and shape have been implicated in the regulation of mtCa$^{2+}$ uptake. Using MFN2 KO or a DRP1 dominant negative mutation, Kowaltowski and colleagues [35] described significant effects of these proteins: They are partially responsible for the effect of mitochondrial morphology on mitoCa$^{2+}$ uptake but do not affect MCU expression. MFN2 KO decreases both mitoCa$^{2+}$ uptake and Ca$^{2+}$ retention capacity in mitochondria. DRP1 dominant negative mutation enhances both mitoCa$^{2+}$ uptake capacity and rates. In mice, constitutive DRP1 ablation reduces muscle growth [36]. In the present study, MICU2 KO did not affect the number of mitochondria, but the phospho-DRP1/DRP1 ratio was strongly increased and associated with a higher fragmentation level of mitochondria and reduced mitoCa$^{2+}$ uptake. We demonstrated that MICU2 is an important actor for the regulation of mitoCa$^{2+}$ uptake and mitochondrial morphology by altering phospho-DRP1/DRP1 activity.

## MICU2 promotes the functioning of the respiratory chain

MICU1 and MICU2 KO cells exhibit altered [Ca$^{2+}$] threshold for mitoCa$^{2+}$ uptake [14]. MICU2 acts as the genuine gatekeeper of MCU at low cytosolic [Ca$^{2+}$],[15] despite the huge driving force for matrix cation entry, thus preventing mitoCa$^{2+}$ accumulation and Ca$^{2+}$ overload [37]. MitoCa$^{2+}$ acts as a key regulator of oxidative metabolism in mammalian cells [38]. In cancer cell lines, constitutive Ca$^{2+}$ flow from ER to mitochondria regulates OXPHOS activity [39]. In permeabilized cells, we mimicked cytosolic [Ca$^{2+}$] ranging from 0 nM to 8.4 μm. Interestingly, MICU2 was involved in the proper functioning of the respiratory chain and the production of ATP in the absence of cytosolic Ca$^{2+}$ or at a physiological concentration of 150 nM. Surprisingly, we demonstrated that loss of MICU2 expression results in a specific defect in the activity and expression of complex IV (cytochrome *c* oxidase), limiting supercomplex assembly and finally maximal OXPHOS capacity, as determined on permeabilized cells, even in absence of cytosolic Ca$^{2+}$. Paupe and colleagues [40] also reported the link between mitoCa$^{2+}$ uptake and defective complex IV assembly. The suppression of CCDC90A, also named MCUR1, in human fibroblasts produces a specific cytochrome *c* oxidase assembly defect, resulting in decreased mitoCa$^{2+}$ uptake capacity. The consequences of MICU2 KO on ATP production in Ca$^{2+}$-free conditions, and its effect on complex IV activity, may also point to a

role for MICU2 in the respiratory chain, independent of $Ca^{2+}$ and possibly independent of MCU. Additional experiments are required to determine this possible role.

## The pivotal role of MICU2 in the metabolic orientation

Mitochondria are the main mediators for the change in metabolic activity of cancer cells [41]. This metabolic reprogramming is implicated in cancer progression and therapy resistance [42].

*CRC* mostly arises from progressive accumulation of somatic mutations within cells. APC, KRAS, p53, and MYC have been reported to participate in genetically global metabolic reprogramming [43]. Pyruvate, produce through glycolysis, glutamine, and fatty acid metabolism form the substrates of the TCA cycle. All these pathways are altered in CRC [44]. Glucose deprivation contributes to the development of KRAS pathway mutations in tumor cells [45]. Oncogenic KRAS decouples glucose and glutamine metabolism to support cancer cell growth [46,47]. p53 regulates biosynthesis through direct inactivation of glucose-6-phosphate dehydrogenase [48]. $MitoCa^{2+}$ uptake is essential to activate $Ca^{2+}$-dependent dehydrogenases of the TCA cycle and to increase OXPHOS. Our results are consistent with the observations of Kaldma and colleagues [49], who conducted an in situ study demonstrating that the OXPHOS system is a major provider of ATP in CRC cells. They found that inhibition of OXPHOS can suppress CRC proliferation and metastasis. In CRC cells, MCU expression promotes cell proliferation in vitro and in vivo by up-regulating $mitoCa^{2+}$ uptake and energy metabolism [50]. Using pharmacological blockers of the glutamine pathway or cell culture media devoid of glutamine, we observed that this substrate is essential for the viability of CRC cells. Furthermore, the oxidative metabolism and mitochondrial ATP production sustain through glutaminolysis is reduced by the down-expression of MICU2.

Here, we also described increased lipid accumulation in MICU2 KO cells and a reduction in the use of beta-oxidation related to proliferation mechanisms. Wang and colleagues [51] showed that CPT1A-mediated activation of fatty acid oxidation increases the metastatic capacity. We also reported that blocking CPT1 and beta-oxidation with etomoxir and trimetazidine reduced the viability of Control KO cells but did not affect the viability of MICU2 KO cells. Recently, Tomar and colleagues [52] demonstrated that MCU and $mitoCa^{2+}$ shape bioenergetics and lipid homeostasis in hepatic cells.

Glucose is the primary source of energy of the cells and it supports important metabolic intermediates, and this contribution is thought to exert great effects on tumor cell metabolism. A decrease in mitochondrial ATP production implies an increase in energy synthesis by glycolysis. Because of their very different efficiencies, this means an increase in glucose consumption. In this study, we showed an increase in glucose uptake in MICU2 KO CRC cells. Furthermore, 2DG treatment led to a greater reduction in the viability of MICU2 KO cells than Control KO cells. These results corroborate the increase in lactate production in MICU2 KO cells, suggesting more glycolytic metabolism in MICU2 KO cells. Interestingly, inhibition of MPC with UK5099 did not affect the viability of MICU2 KO cells. Thus, in the absence of MICU2, cells develop a preference for glycolysis as the main energy supply. $MitoCa^{2+}$ uptake is essential to activate $Ca^{2+}$-dependent dehydrogenases of the TCA cycle and to increase OXPHOS. We demonstrated that MICU2 KO decreased $mitoCa^{2+}$ influx in CRC cells; this modification inhibited the phosphorylation of PDH and the amount of acetyl-CoA.

The role of different MICU isoforms on bioenergetic preference is not very clear. Chakraborty and colleagues [32] recently demonstrated that MICU1 overexpression in ovarian cancer cells induces both glycolysis and chemoresistance and that MICU1 overexpression in patients with ovarian cancer correlates with poor overall survival. In addition, MICU1 negatively regulates OXPHOS function. In the vast majority of cases, cancer cells undergo metabolic

reprogramming and preferentially use glucose via aerobic glycolysis [53]. In contrast, we have defined a role of MICU2 in maintaining an OXPHOS phenotype and the respiratory complex assembly in CRC cells at an advanced stage. Taken together, we propose that association between MICU1/MICU1, MICU2/MICU1, and MICU3/MICU1 implicated in MCU regulation shape the metabolic orientation of cancer cells. Thus, high MICU1 expression is associated with a glycolytic phenotype, while high expression of MICU2 favors respiration and mitochondrial metabolism. However, an MCU-independent contribution of MICU2 cannot be ruled out. Indeed, a recent report described MICU1 as an intermembrane space $Ca^{2+}$ sensor that modulates mitochondrial membrane dynamics independently of matrix $Ca^{2+}$ uptake [54]. Therefore, it is possible that MICU2 exerts a similar role in specific circumstances.

In summary, we have demonstrated a specific role for the MICU2 in the regulation of mitoCa$^{2+}$ uptake in CRC. Given the established role of MICU2 in regulating mitochondrial calcium uptake, mimicking the consequences of MICU2 knockout could provide valuable insights into tumor biology. This approach invites consideration of various known chemical modulators of mitochondrial calcium uptake, such as MCU blockers like mitoxantrone or Benzothenium chloride, and MCU/MICU1 modulators including the MCU-I4 or MCU-I11 inhibitors identified by Di Marco and colleagues [55]. While these agents offer promising tools for modulating mitochondrial calcium levels, it is noteworthy that there is currently no pharmacological inhibitor capable of directly targeting MICU2 to prevent its activation or conformational changes. Addressing this gap could open new therapeutic strategies for reducing tumor proliferation and metastasis, particularly in CRC. By exploring these chemical modulators and developing novel inhibitors that specifically target MICU2, we could uncover new mechanisms for controlling mitochondrial calcium uptake, potentially leading to effective interventions against CRC and other malignancies where mitochondrial calcium homeostasis plays a critical role in tumor progression.

We have also defined a crucial role for MICU2 in maintaining the integrity of the mitochondrial network and in the proper assembly and functioning of the respiratory chain. For the first time, we have demonstrated the central role of MICU2 in the balance between glycolysis and OXPHOS during cancer progression, and more specifically in CRC.

This work provides a promising perspective to better understand and target metabolic diseases including cancers. The MICU2/MICU1 ratio could represent a singular predictive marker of the metabolic preferences and pave the way for the development of personalized metabolic therapies.

## Methods

### Transcriptomic and proteomic analysis

Transcriptomic analyses and heatmaps were generated using R software. RNA-Seq data from colon cancer and normal colon were generated by The Cancer Genome Atlas Research Network (http://cancergenome.nih.gov/). Read counts and clinical annotation were downloaded from GDC using the *tcgaworkflow* package. Read counts were then filtered, normalized, processed, and logarithmically transformed using the *edgeR* package. Primary tumors of TCGA--COAD were classified as "High" or "Low" if the value of their expression is respectively superior to the 95th percentile or inferior to the 5th percentile of their expression in normal samples. Differential expression analysis was performed on read counts using the edgeR package. DEA was obtained by comparing groups defined according to the expression of MICU2, MICU1, or the MICU2/MICU1 ratio. Heatmaps of the expression of top differentially genes (Fig 2A) of genes associated to hypoxia, glycolysis, mitochondrial respiration or TCA cycle (Fig 6J) were generated using the *ComplexHeatmap* package [56]. GSEA was performed on the

log$_2$(fold change) ranked gene lists obtained by DEA using the *fgsea* package and gene set collections from MSigDB (https://www.gsea-msigdb.org/gsea/msigdb-c5.go.bp.v7.5.1.symbols.gmt). Published transcriptomic data of normal, primary, and metastatic CRC generated by microarrays (GSE41258) were used [57]. Raw data were downloaded and RMA preprocessed using the *affy* package. Proteomics data were obtained from CPTAC data portal and from Zhang and colleagues [17]. Tumor samples barcodes were used to retrieve clinical data from TCGA-COAD and TCGA-READ cohorts. Kaplan–Meier plot and survival analysis were performed using *survminer* and *survival* R packages. High and Low group for survival analysis were define based on the median of protein abundance of protein of interest. *P*-value of the difference between High and Low groups on the overall survival were calculated using Gehan–Breslow method.

## Cell culture

The CRC cell lines HT29, DLD1, HCT116, SW480, and SW620 were chosen to investigate the role of MICU2. CRC cells were cultured in RPMI-1640 media + GlutaMax (Gibco, Thermo Fisher, Illkirch, France) supplemented with 10% fetal bovine serum (FBS; Eurobio, Les Ulis, France) and 1% penicillin and streptomycin (Eurobio). NCM356 cells (Incell Corporation, LLC, San Antonio, Texas, USA) were cultured in high-glucose Dulbecco's Modified Eagle Medium (DMEM) (Sigma-Aldrich, Missouri, USA) supplemented with 10% FBS and antibiotics. All cells were cultured in an incubator at 37°C with 5% CO$_2$.

## Generation MICU2 KO

Several MICU2 KO clones in HCT116-Luc cells were generated by Ubigene (China) using the CRISPR/Cas9 system. MICU2 KO was generated using 2 exon 2 targeting sequences, MICU2-gRNA3 (ACACTTAGAGATTAAACGAGG) and MICU2 gRNA6 (GTATTCCAGTACACTTAGGAAGG). All clones were validated by polymerase chain reaction (PCR) and imaging. The HCT116 MCU KO line was obtained and validated by Pr Trebak's laboratory [58].

## Metabolic pathway inhibitors

The following inhibitor references are described in the key resources table: 2DG (5 mM), 2NBDG (100 μm), UK5099 (50 μm), rotenone (0.5, 1, and 2.5 μm), oligomycin (0.5, 1, and 2.5 μm), trimetazidine (10 μm), etomoxir (10 μm), telaglenastat (1 and 5 μm), V-9302 (5 and 10 μm), BAPTA (2.5 and 5 μm), and MITOTEMPO (2.5 and 5 μm).

## Specific media

Specific media were used: RPMI Medium 1640 [–] Glutamine (Reference: 42401018, Gibco, Thermo Fisher, Illkirch, France), RPMI Medium 1640 [–] Glucose (Reference: 11879020, Gibco, Thermo Fisher, Illkirch, France), RPMI Medium 1640 [+] 4.5 g/L D-Glucose (Reference: A1049101, Gibco, Thermo Fisher, Illkirch, France).

## Cell proliferation

After harvesting, $3 \times 10^3$–$5 \times 10^3$ cells were plated in each well of 96-well plates. The platers were incubated at 37°C in 5% CO$_2$ for 4 h to allow the cells to adhere to the plate. The Cyquant-NF dye was diluted in HBSS buffer, and 100 μl of the mixture was added in each well. The plate was kept at 37°C in 5% CO$_2$. The fluorescence intensity (~485/~530 nm) was measured using FlexStation 3 Multimode Plate Reader (Molecular Devices, California, USA).

## Cell viability

Cell viability, survival in HBSS, and short-term toxicity were evaluated using standard sulfor-hodamine B (SRB) method after treatment for 24 and 48 h. Briefly, cells were fixed with 50% trichloroacetic acid for 1 h at 4°C and stained for 15 min with 0.4% SRB solution. Cells were then washed 3 times with 1% acetic acid and dye was dissolved using 10 mM Tris over 10 min. Absorbance at 540 nm was determined with a BioTek (Vermont, United States) Spectrophotometer.

## Cell cycle

Prior to staining, cells were washed twice by centrifugation in phosphate-buffered saline (PBS) at 500 g for 5 min. Next, 106 cells were permeabilized with 1 ml ice-cold ethanol (1 h, 4°C). After 2 washes with PBS, 1% FBS, and 0. 25% Triton X-100 (PFT), cells were stained in 200 µl PFT for 30 min at room temperature in the dark with 10 µg 7-AAD (Sigma-Aldrich), 5 µl Alexa Fluor488-conjugated anti-human Ki67 mAb (B56) (Becton-Dickinson, Pont-de-Claix, France) and 3 µl Alexa Fluor488-conjugated polyclonal anti-phospho(ser10)-histone H3 antibody (Cell Signaling Technology, Danvers, Massachusetts, USA). A control tube was prepared with 10 µg 7-AAD and 5 µl Alexa Fluor488-conjugated mouse IgG1 (Becton-Dickinson). After 2 washes with PFT, cells were stained with 10 µl of anti-CD45 (A20) antibody conjugated to APC-Cy7 or with 5 µl of anti-CD3 (UCHT1) antibody conjugated to Becton-Dickinson's Horizon V450, followed by incubation for 20 min at 4°C. Cells were then washed twice with PBS, centrifuged for 5 min at 500 g and resuspended in 300 µl of PBS. Samples were analyzed on a FACSCanto II flow cytometer (Becton-Dickinson) equipped with 3 lasers, 1 blue (488-nm, air-cooled, 20-mW solid state), 1 red (633-nm, 17-mW HeNe), and 1 violet (405-nm, 30-mW solid state). Green fluorescence (Alexa Fluor488 emission) was collected after passing through a 530/30 nm band-pass filter (BP). APC-Cy7 emission was detected by filtration through a 780/60 nm BP filter and Horizon V450 emission by filtration through a 450/50 nm BP filter; 7-AAD emission was collected after passing through a 650 nm long-pass filter [59].

## Cell migration in the wound healing assay

First, $6 \times 10^5$ cells were seeded on each side of an Ibidi culture insert (Ibidi GmbH, Gräfelfing, Germany) in 100 µl of RPMI supplemented with 10% FBS and enriched with 1% antibiotics. The 24-well plate was incubated at 37°C and 5% $CO_2$ for 24 h. The insert was then removed to create the scar and 500 µl of medium was added. The plate containing the scarred cells was then placed in a chamber maintained at 37°C and 5% $CO_2$, connected to a camera so that photographs of the scar could be taken at regular intervals for 24 h. At the end of the migration, the area of the scar not covered by cells was measured using the T-scratch software. The percentage of coverage is calculated using the following formula:

$$\text{Recovery \%} = \frac{AT0H - AT24H}{AT0H},$$

where $A_{T\ 0H}$ is the uncovered scar area measured at time T0 immediately after scar creation and $A_{T\ 24H}$ is the uncovered scar area measured at time T24, 24 h after scar creation.

## Spheroid base migration

Spheroids were prepared by seeding drops of cells at a rate of $3 \times 10^3$ cells per 30 µl into the lid of a petri dish (Reference: 353003, Falcon), the lid was then turned over to form a spheroid. After 4 days, the spheroid was placed in a 96-well plate (Reference: 353077, Falcon) coated

with fibronectin and the migration around the spheroid was photographed using a microscope (Nikon, Champigny-sur-Marne, France) at 0, 24, and 48 h.

## Subcutaneous CRC xenograft tumor model

CRC xenografts were established by injecting $5 \times 10^6$ cells per Swiss nude mouse ($n = 10$ per cell line) as described previously by Gueguinou and colleagues [60]. All mice were assessed weekly using whole-body bioluminescence imaging and photoacoustic imaging as described below.

## Ultrasound/photoacoustic imaging, bioluminescence imaging, and hypoxia immunostaining

The protocols have already been described in detail [61]. Briefly, bioluminescence imaging was performed once a week until the end of the study using an IVIS-Lumina II (Perkin Elmer, France). Each mouse was injected intraperitoneally with 100 mg/kg luciferin potassium salt (Promega, France). After mice were anesthetized, acquisition binning and duration were set depending on tumor activity. Signal intensity was quantified as the total flux (photons/seconds) within regions of interest drawn manually around the tumor area using Living Image 4.4 software (Perkin Elmer).

For photoacoustic imaging, mice were anesthetized with 1.5% isoflurane and placed on a thermostatically heating pad (37°C). Respiratory gating was derived from electrocardiography. A colorless aqueous warmed ultrasonic gel (Supragel1, LCH, France) without any air bubbles was applied between the skin and the transducer. Tumors were imaged with the VisualSonics VevoLAZR System (FUJIFILM VisualSonics, Canada). Three-dimensional (3D) scans were made of digitally recorded ultrasound images. The tumor area was measured by delineating the margins using Vevo1LAB 3.2.6 software. For hypoxia assessments, tumors were investigated by photoacoustic imaging with the OxyHemo-Mode to determine the average SO2 values and the corresponding hypoxic volumes. A transducer with central frequency at 21 MHz was used for B-Mode imaging and photoacoustic imaging.

Pimonidazole immunostaining was performed with the Hypoxyprobe kit (Hypoxyprobe, Burlington, USA), following the supplier's instructions. Briefly, mice received an intravenous injection of 60 mg/kg of the pimonidazole solution and were sacrificed 60 min following the injection. The tumors were then resected and fixed for 24 h in 10% formaldehyde. The tumors were embedded in paraffin and sectioned. The primary antibody was mouse anti-pimonidazole and the secondary antibody was FITC-labeled goat anti-mouse (Abcam, Cambridge, USA).

## Metastasis CRC xenograft model

The hepatic metastasis model simulates an advanced stage of CRC by injecting tumor cells into the spleen of anesthetized mice, which then rapidly migrate to the liver, forming metastases without a primary colon tumor. The procedure involves anesthetizing mice with isoflurane, administering preoperative analgesics (lidocaine, bupivacaine, buprenorphine), and performing a small incision to expose the spleen for cell injection. After injection, the spleen is removed, the incision is sutured, and postoperative analgesia (Tolfedine) is given. Mice recover on a heating pad before returning to their home cage.

For monitoring, mice are weighed weekly and bioluminescence imaging is conducted from the seventh day post injection to track metastasis formation and growth, using intraperitoneal luciferin and isoflurane anesthesia during imaging. These procedures ensure minimal handling time. The study uses 8-week-old female BALB/c Nude mice from Charles River, which are acclimated for a week before induction at 9 weeks.

## Animal ethics

The project titled "Experimental Cancer Models Using Colorectal Carcinoma Cells for the Evaluation of New Therapeutics via In Vivo Imaging," referenced as APAFIS (Animals used for Scientific Purposes) #38405–2022011116011226 v4, has been authorized for a period of 5 years from September 14, 2022. Submitted by CNRS UPS 44 TAAM - Orléans (accreditation number D452346), the project was ethically reviewed by the Animal Experimentation Ethics Committee No. 003 and received a favorable opinion. The authorization is contingent on the continued validity of the establishment's accreditation and is not subject to mandatory retrospective assessment. This study was performed in strict accordance with the recommendations in the Guide for the Care and Use of Laboratory Animals. Every effort was made to minimize animal suffering.

## Lipid droplet staining

Forty-eight hours before fixation, $1 \times 10^5$ cells were placed per well in an 8-well cell culture chamber (Reference: 154534, Labtek, Brendale, QLD). The cells were fixed with a 4% paraformaldehyde at room temperature for 15 min. Cell culture chambers were rinsed twice with cold PBS, permeabilized with 0.5% Triton X-100 for 5 min at room temperature, rinsed 3 times with PBS, and then incubated with blocking solution (PBS containing 0.2% bovine serum albumin [BSA], 0.02% sodium azide, 0.05% Triton X-100, and 10% FBS) for 1 h at room temperature. Cells were stained with 0.3% Oil Red O (diluted in 60% isopropanol) for 2 min at room temperature. The cell culture chambers were then washed twice in PBS and nuclei were stained with DAPI for 5 min. The cell culture chambers were then washed once more and mounted with coverslips using SlowFade Gold antifade mounting medium (Reference: P36961, Molecular Probes, Oregon, USA).

Oil Red O was used to quantify the number of lipid droplets. The images were acquired with a Leica SP8 microscope (microscopy department of the University of Tours). Image processing and analysis were performed using ImageJ/Fiji software (National Institutes of Health, Bethesda, Maryland, USA). After subtracting the background (using a 20-pixel rolling ball radius) and setting an identical threshold for all images, the particle analysis function was used to measure the number of lipid droplets per cell.

## Immunohistochemistry

Immunohistochemistry was done with a Discovery Ultra Machine (Roche, Mannheim, Germany) and the Chromo-Map Kit (Reference: 760–159, Roche). The primary antibody (Ki67) was diluted with Antibody Diluent (Reference: 760–108, Roche) and the samples were incubated 60 min at 37˚C. Automatic application of secondary antibody was done with Omni Map anti-Rb HRP (Reference: 760–4311, Roche). The staining was visualized by automatic addition of the substrate ($H_2O_2$+DAB) of the Chromo-Map Kit and the HRP of the secondary antibody. Counter-staining was performed with Hematoxylin II (Reference: 790–2208, Roche) for 16 min and Bluing Reagent (Reference: 760–2037, Roche) for 4 min. The slides were then dehydrated and mounted with Pertex mounting media (Reference: 00811-EX, Histolab, Askim).

## 3D fluorescence microscopy

Cells were transfected with a mitochondria-targeted fluorescent protein (PDH_GFP) or incubated for 15 min with 100 nM Mitotracker green (Molecular Probes) to stain the mitochondrial network. For fluorescence imaging, coverslips were mounted in housing and placed on the stage of an inverted wide-field ECLIPSE Ti-E microscope (Nikon, Tokyo, Japan) equipped

with a 100× oil immersion objective lens (Nikon Plan Apo100x, N.A. 1.45) and an Andor NEO sCOMS camera controlled by Metamorph 7.7 software (Molecular Devices, Sunnyvale, California, USA). A precision piezoelectric driver mounted underneath the objective lens allowed faster Z-step movements, keeping the sample immobile while shifting the objective lens. Fifty-five image planes were acquired along the Z-axis at 0.1-μm increments. For mitochondrial network characterization, the acquired images were iteratively deconvolved using Huygens Essential software (Scientific Volume Imaging, Hilversum, the Netherlands), with a maximum iteration score of 50 and a quality threshold of 0.01. Imaris 8.0 software (Bitplane, Zurich, Switzerland) was used for 3D processing and morphometric analysis. The mitochondrial network was modeled in 3D, and thresholds were defined to classify mitochondria depending on their volume (Imaris Isosurface Tools).

## Reverse transcription real-time polymerase chain reaction (RT-qPCR)

Total RNA was collected using the Nucleospin RNA Kit (Macherey–Nagel, Hoerdt, France) and transcribed into complementary DNA (cDNA) with the PrimeScript RT Reagent Kit (RR037A, Takara, Kusatsu, Japan). cDNA was then amplified with the SYBR Green Master kit (Roche) using a Light Cycler 480 apparatus. RT-qPCR was performed in 40 cycles of 95˚C for 15 s and 60˚C for 45 s. The average ΔCt value was calculated for each cell line with respect to the housekeeping gene HPRT1. The primers sequences used are listed in S1 Table.

## Western blot analysis

HT29, DLD1, SW480, HCT116, and SW620 colon cancer cells and NCM356 non-tumoral cells were collected with a lysis buffer containing protease inhibitors and phosphatase inhibitors and used for protein assay. A BCA protein assay kit (Reference: 23227, Thermo Fisher Scientific, France) was used to determine the protein concentration. The resulting proteins were separated by polyacrylamide gel electrophoresis (4% to 15% Mini-PROTEAN TGX Stain-Free Protein Gels, Bio-Rad, California, USA) and then transferred to a nitrocellulose membrane (Reference: 1704158, Bio-Rad). The membranes were incubated in primary antibody diluted 1:1,000 in TBST with 5% milk overnight at 4˚C with agitation. The primary antibodies were against MCU (Sigma-Aldrich), MICU2 (Abcam), MICU1 (Sigma-Aldrich), MFN2 (Cell Signaling, USA), OPA1 (BD Biosciences, France), DRP1 (Cell Signaling), PDRP1 (Cell Signaling). The following day, the membranes were washed 3 times with TBST and incubated with HRP-conjugated secondary antibody at room temperature for 1 h with shaking. The secondary antibodies used were: m-IgG Fc BP-HRP (1:5,000, sc-525409) and Ms x Rb Light Chain specific HRP (1:2,000, MAB201P). The internal control was total protein deposited and quantified with a gel stain. An ECL chemiluminescence kit (Clarity Western ECL substrate, Bio-Rad) was used to visualize the bands, which were quantified with Image Lab software (Bio-Rad).

## MitoCa$^{2+}$ measurements

To measure mitoCa$^{2+}$ using mito-SP-linker-GCaMP6m, named mt-riG6m, cultured cells were transfected using Lipofectamine 2000 (Invitrogen, USA) with 1.5 μg of pCDNA3.1 mt-riG6m 2 days before imaging. Twenty-four hours before the experiments, transfected cells were plated in FluoroDish FD35-100. The cells were stimulated with 10 μm LPA in PBS containing 2 mM CaCl$_2$. Time-lapse images were acquired using epifluorescence microscopy (Nikon). mt-riG6m and miGer were gifts from Li and colleagues [62].

ER Ca$^{2+}$ store depletion was quantified by transfecting parental and MICU2-KO HCT116 cells with red miGer [62] using Lipofectamine 48 h prior to imaging. Cells expressing miGer were then excited at 552 nm and relative ER Ca$^{2+}$ measurements were recorded through a 20×

objective lens. Immediately after identifying R-miGer-positive cells, the bath solution was replaced with nominally $Ca^{2+}$-free HBSS. One minute into the experiment, the cells were treated with TG (Reference: T7458, Life Technologies, Thermo Fisher).

## SOCE measurement by Fura-2 AM

Intracellular $Ca^{2+}$ imaging was performed as described previously [63,64]. Cells were plated in 96-well plates at $2 \times 10^4$ cells per well 24 h before the experiment. Adherent cells were for loaded with the ratiometric dye Fura2-acetoxymethyl ester (AM; 2 μm) at 37˚C for 45 min and then washed with PBS supplemented with $Ca^{2+}$. During the experiment, the cells were incubated with $Ca^{2+}$-free physiological saline solution (PSS) solution and treated with 2 μm TG to deplete intracellular store of $Ca^{2+}$. $Ca^{2+}$ entry was stimulated by injecting 2 mM of $CaCl_2$. Fluorescence emission was measured at 510 nm using the FlexStation-3 (Molecular Devices) with excitation at 340 and 380 nm. The maximum fluorescence (peak of $Ca^{2+}$ influx [F340/F380]) was measured in MICU2 KO cells and compared with the Control cells.

## Measurement of oxygen consumption

**Seahorse analysis.** The cellular oxygen consumption rate (OCR) data were obtained using a Seahorse XF96 Flux Analyzer from Seahorse Bioscience (Agilent Technologies, Santa Clara, California, USA). The experiments were performed according to the manufacturer's instructions. Briefly, HCT116 cells were seeded in XF96 cell culture plates at $2 \times 10^4$ cells/well. On the day of analysis, the culture medium was replaced with XF DMEM (Thermo Fisher Scientific, San Jose, California, USA) supplemented with 2 mM glutamine and lacking bicarbonate (pH 7.4). The cells were then incubated at 37˚C in a non-$CO_2$ incubator for 1 h. Measurements were made as described in the relevant figure legends. Sequential injection of 10 mM glucose, 1 μm oligomycin, 100 μm dinitrophenol (DNP), and 0.5 μm rotenone/antimycin A permitted the determination of the main respiratory parameters. Finally, the data were normalized to the amount of DNA present in the cells and assayed using the Cyquant Cell Proliferation Assay kit (Thermo Fisher Scientific, San Jose, California, USA). The data were acquired with the Seahorse Wave Controller and analyzed with the Seahorse Wave Desktop Software.

**Whole cell measurement.** The measurements were performed at 37˚C under magnetic stirring by placing $2 \times 10^6$ cells in a final volume of 2 ml of culture medium (RPMI, RPMI without glucose, or RPMI with high glucose) with 10% FBS in an oxygraph (OROBOROS oxygraph-2k). The cells of each line were treated with their respective medium 48 h before the experiment. They were then harvested, counted, and centrifuged at 700$g$ for 3 min. The cell pellet was suspended in 5 ml of selected medium with FBS and centrifuged at 800$g$ for 2 min. The resulting residual pellet was then resuspended in 500 μl of selected medium with SVF (cell solution used for the oxygenation measurement). The resuspended cells were added to 1.5 ml of different medium with FBS already present in the measuring tank. As soon as a steady slope was obtained, the oxygen consumption was measured. OXPHOS was then inhibited by adding oligomycin (an ATP synthase inhibitor; 5.25 μg/ml); the respiration rate obtained corresponds the energy wasting state. Successive additions of FCCP (a protonophoric uncoupler; 0.8 μm) were then made until the maximum respiration rate was reached. Finally, 2 μm antimycin A was added to determine non-mitochondrial oxygen consumption.

**Measurement of permeabilized cells.** Measurements were performed at 37˚C—with agitation, in a final volume of 2 ml of respiration buffer with BSA and different [$Ca^{2+}$] (0, 500 nM, 1 μm and 8.4 μm) with $3 \times 10^6$ cells (OROBOROS oxygraph-2k) previously permeabilized with digitonin. The cells of each line were detached with trypsin, counted and centrifuged at

700$g$ for 3 min. The pellet was suspended in 1 ml of ASB-free respiration buffer and the plasma membrane was permeabilized by adding digitonin (1 mg/ml previously heated at 95˚C for 2.5 min) (6 µl per $1 \times 10^6$ HCT116 cells). The whole set was gently shaken for 2 min. Then, 4 ml of respiration buffer (10 mM $KH_2PO_4$, 300 mM mannitol, 10 mM KCl, 5 mM $MgCl_2$, and 1 mM EGTA, pH 7.4 at 37˚C) with ASB fatty acid free was added to stop the action of digitonin. The mixture was centrifuged at 800$g$ for 2 min. The resulting pellet was then resuspended in 500 µl of respiration buffer with BSA (solution for oxygen measurement). The cells resuspended in the respiration buffer with BSA were added to the measuring cell containing 1.5 ml of respiration buffer with BSA. As soon as a steady slope was obtained, succinate (10 mM) and ADP (1.5 mM) were added and the oxygen consumption related to ATP synthesis was determined. OXPHOS was then inhibited by adding oligomycin (5.25 µg/ml). Once the oxygen consumption stabilized, FCCP (0.8 µm) was added until the maximum respiration was reached. Cytochrome $c$ was added, to control the integrity of the outer mitochondrial membrane and then 2 µm antimycin A was added to inhibit complex III.

## Expression of super complexes and MCU complexes

For supercomplex assembly analyses, mitochondria were enriched by using differential centrifugation, according to a slightly modified version of the method described by Bonnet and colleagues [65]. Briefly, the cell pellets were incubated for 10 min on ice with cold digitonin (4 mg/ml in PBS [v/v] or 0.2% w/v) to dissolve cell membranes. Then, digitonin was diluted by adding cold PBS (5% v/v). Cells were centrifugated at 10,000$g$ for 10 min at 4˚C to recover the mitochondria-enriched fraction. The pellet was washed once more in 1 ml of cold PBS, centrifuged (10,000$g$ at 4˚C), resuspended at 10 mg/ml in AC/BT buffer (1.5M aminocaproic acid and 75 mM Bis-Tris/HCl (pH 7.0), supplemented with Complete Mini Protease Inhibitor [Roche Diagnostics, Stockholm, Sweden]), and kept frozen at −80˚C until analysis.

For blue native polyacrylamide gel electrophoresis (BN-PAGE) analyses, 30 to 50 µg (MCU complex and respiratory chain supercomplexes analyses, respectively) of each sample was diluted at 2 mg/ml in AC/BT buffer and mitochondrial membrane proteins were solubilized by incubation with 6g/g protein digitonin (1.2% w/v) for 10 min at 4˚C. After centrifugation at 20,000$g$ for 20 min at 4˚C, the supernatant was collected, and 5% Serva Blue G dye (Bio-Rad) in 1 M aminocaproic acid/50 mM Bis–Tris/HCl (pH 7.0) was added (1/20 v/v) prior to loading. MCU complex, respiratory chain complexes and supercomplexes were separated on Native PAGE Novex 3% to 12% Bis-Tris gels (Invitrogen) for approximately 3 h and transferred to polyvinylidene fluoride membranes (GE Healthcare, Velizy-Villacoublay, France) in cold blotting buffer (25 mM Tris, 192 mM glycine (pH 8.3), and 20% methanol). For respiratory chain supercomplexes analyses, membranes were hybridized using dedicated monoclonal antibodies (NDUFS2 rabbit monoclonal (AB192022), SDHA rabbit monoclonal (AB137040), UQCRC2 mouse monoclonal (AB14745), MT-CO1 (COX1, AB14705) mouse monoclonal, Abcam, the Netherlands). Secondary antibodies coupled to Alexa Fluor infra-red fluorescence (AB186697 and AB186696) were used. MCU complex was hybridized using anti-MCU rabbit monoclonal antibody (14997, Cell Signaling Technology) and SDHA as charge reference. Goat anti-Rabbit secondary antibody coupled to HRP (A27036, Invitrogen) was detected by chemiluminescence (Thermo Fischer Scientific, 34095). Acquisitions were performed using the Odyssey FC imaging system and analyzed using Image Studio Lite (LI-COR Biosciences).

## Mitochondrial enzymatic activities

The activities of complex I (NADH ubiquinone reductase) and citrate synthase (CS) were measured at 37˚C with a UVmc2 spectrophotometer (SAFAS, Monaco) in a mitochondria-

enriched fraction. Cells were resuspended in cell buffer (250 mM saccharose, 20 mM tris [hydroxymethyl]aminomethane, 2 mM EGTA, 1 mg/ml BSA (pH 7.2); $1 \times 10^6$ cells per 50 μl), disrupted by 2 freeze–thaw cycles, washed, and centrifuged at 16,000$g$ for 1 min to eliminate the cytosolic fraction. For complex I and V analyses, cells were further resuspended in the cell buffer ($1 \times 10^6$ cells per 250 μl) and sonicated (6 times × 5 s) on ice. The activities of respiratory chain complex activities were measured according to standard routine clinical protocols for CI (NADH:ubiquinone reductase, NUR) [66], complex II (succinate:ubiquinone reductase, SUR), Complex III (ubiquinol:cytochrome $c$ reductase, UCCR), Complex IV (cytochrome $c$ oxidase, COX) [67], Complex V (F1-ATPase), and CS [67]. The protein content was determined with the bicinchoninic assay kit (Uptima, Interchim, Montluçon, France) using BSA as the standard.

## PDH activity assays

The PDH activity of MICU2 KO cells was measured with the colorimetric PDH Assay Kit (Sigma-Aldrich) according to the manufacturer's instructions.

## Acetyl-CoA ratio measurement assay

The acetyl-CoA content of the cell culture medium was determined with the Acetyl-CoA Assay Kit (Reference: ab87546, Abcam) according to the manufacturer's instructions.

## Lactate production

Lactate production was assessed with the L-Lactic Acid Colorimetric Assay Kit (Reference: E-BC-K044-S, Elabscience, Texas, USA) according to the manufacturer's instructions.

## Fatty acid oxidation activity

Visualization of fatty acid oxidation in the HCT116 Control and KO MICU2 cells lines treated with FAOBlue (Funakoshi, Japan) in HEPES-buffered saline (HBS) buffer with or without pre-treatment with etomoxir (20 μm, 3 h), a potent fatty acid oxidation inhibitor. After FAO-Blue incubation, blue fluorescence (excitation 405 nm, emission 430 to 480nm) was observed.

## Detection of ROS

A total of $1 \times 10^6$ cells were stained for 10 min at 37˚C with 5 μm DCFDA and then washed and analyzed by flow cytometry (excitation approximately 492 to 495 nm, emission 517 to 527 nm) to detect cytosolic ROS. Mitochondria-derived ROS were detected in $1 \times 10^6$ cells stained at 37˚C for 10 min with 5 μm MitoSOX (DHE). After staining, the cells were washed and analyzed by flow cytometry (excitation 510 nm, emission 580 nm). The cells were analyzed immediately after completing staining.

## Statistical analysis

The data are presented as the mean ± standard error of the mean and were analyzed with GraphPad Prism 6 (GraphPad Software, San Diego, California, USA). Statistical analysis was performed with a paired Mann–Whitney test, the Kruskal–Wallis test, or analysis of variance (ANOVA) followed by Dunnett's, Dunn's, or Sidak's multiple comparisons test. The exact tests are indicated in the figure legends. A $p$-value $<0.05$ was considered significant.

## Supporting information

**S1 Fig. MICU2 KO and expression of MCU, MICU1, and MICU2/MICU1 ratio in CRC cells.** (A) Transcriptomic analysis of the MICU2/MICU1 ratio according to CRC stages in the TCGA-COAD (B) Transcriptomic analysis of the MICU1 and MICU2/MICU1 ratio in the GSE41258 data sets. Each data point represents an individual sample (ANOVA followed by Dunn's multiple comparisons test). (C) Bar plot representing the expression of MCU, MICU1, MICU1 variant, and MICU2 mRNAs measured by RT-qPCR in the HCT116 Control and MICU2 KO cell lines ($n = 3$, ANOVA followed Dunnett's multiple comparisons test, $**p < 0.01$ and $***p < 0.001$). (D) Density plots representing the definition of the status of MICU1, MICU2, and the MICU2/MICU1 ratio of primary colon tumor samples of the TCGA-COAD data set. (E) Proteomics analysis generated by CPTAC consortium of 95 tumor samples included into the TCGA-COAD and TCGA-READ genomic projects. (Upper left) Violin plots representing the relative protein abundance of MCU, MICU1, MICU2, and EMRE (SMDT1) in tumor samples in function of AJCC pathological stages (Stages I-II-III compared to Stage IV, upper left), of the presence of a vascular invasion (upper right), of the presence of distant metastasis (lower left). Kaplan–Meier survival plot representing the overall survival probability of patients according the levels of expression of MCU, MICU1, MICU2, and EMRE (SMDT1). High and Low groups have been determined based on the median of protein abundance of individual protein in tumor samples. The data underlying the graphs shown in the figure can be found in S1 and S2 Data.
(TIF)

**S2 Fig. The role of MICU2 in cell proliferation, metastasis formation, and sensitivity to chemotherapy.** (A) Heatmap of the most differentially expressed genes between colon tumor samples in the TCGA-COAD data set with low, normal, or high expression of MICU2/MICU1 ratio. (B) Heatmap of normalized enrichment scores of all cell cycle–associated genes obtained by GSEA for primary colon tumor samples with low, normal, or high expression of MICU2. A star indicates an adjusted $p$-value $<0.05$. (C) Boxplots representing the viability of the Control and MICU2 KO cell lines cultured in the absence of presence in 5FU (500 nM, 1 μm, 5 μm, and 10 μm) (top panel) and oxaliplatin (500 nM, 1 μm, 5 μm, 10 μm, and 20 μm) (bottom panel) for 24 and 48 h ($n = 9–11$). (D) Left panel: representative images of the wound healing assay. Right panel: curve and boxplots representing the percentage of gap closure as a function of time and after 24 h in the Control and MICU2 KO cell lines. The scale bar is 5 μm ($n = 6–8$, Kruskal–Wallis test). (E) Left panel: representative images of 3D spheroids of the Control and MICU2 KO cell lines plated on top of a fibronectin layer after 0, 24, and 48 h. Right panel: box-plot representing the migration area at 24 and 48 h. The scale bar is 20 μm ($n = 5–8$, Kruskal–Wallis test). (F) Representative bioluminescence images of cancer metastasis in mice injected with the luciferase-expressing Control or MICU2 KO cells lines. (G) Quantification of number of photons/s per mouse at indicated days ($n = 5–8$, Kruskal–Wallis test). (H) Representative image showing the formation of metastatic nodules in the liver of mice injected with HCT116 WT or MICU2 KO cells. On all plots, $*p < 0.05$, $**p < 0.01$, and $***p < 0.001$. The data underlying the graphs shown in the figure can be found in S1, S2 and S4 Data.
(TIF)

**S3 Fig. The effect of MICU2 knockout on mitochondrial dynamics and ROS/Ca$^{2+}$ signaling.** (A) Bar plots representing the fold change of the expression of genes associated with mitochondrial dynamics in the Control and MICU2 KO cell lines compared with the Control KO cell line ($n = 3–5$). (SB) Left panel: representative measurements of cytosolic [Ca$^{2+}$] in the Control and MICU2 KO cell lines measured with Fura-2 AM in response to 1 μm of LPA applied

in the absence of extracellular $Ca^{2+}$ and in the presence of 2 mM of extracellular $Ca^{2+}$. Right panel: boxplot representing the cytosolic Fura-2AM 340/380 ratio measured in the presence of extracellular $Ca^{2+}$ in the Control and MICU2 KO cell lines ($n = 4$, $N = 27–55$). (C) Left panel: representative measurements of cytosolic [$Ca^{2+}$] in the Control and MICU2 KO cell lines measured with Fura-2AM in response to 2 μm TG applied in the absence of extracellular $Ca^{2+}$ and in the presence of 2 mM of extracellular $Ca^{2+}$. Right panel: boxplot representing the basal Fura-2AM 340/380 ratio measured in the presence of extracellular $Ca^{2+}$ in the Control and MICU2 KO cell lines ($n = 6$, $N = 42–95$, ANOVA followed by Dunnett's multiple comparisons test). (D) Bar plots representing the viability of the Control and MICU2 KO cell lines cultured 24 or 48 h in presence or absence of BAPTA-AM (a $Ca^{2+}$ chelator) ($n = 3–5$, ANOVA followed by Dunnett's multiple comparisons test). (E) Boxplots representing ROS production in the Control and MICU2 KO cell lines. Total ROS, mitochondrial ROS, and hydrogen peroxide were measured using DHE, MitoSox, and DCFDA, respectively. (F) Bar plots representing the viability of the Control and MICU2 KO cell lines incubated for 24 h with MITOTEMPO (a mitochondrial ROS chelator) ($n = 5–7$, ANOVA followed by Dunnett's multiple comparisons test). On all plots, $^*p < 0.05$, $^{**}p < 0.01$, and $^{***}p < 0.001$. The data underlying the graphs shown in the figure can be found in S1 Data.
(TIF)

**S4 Fig. Molecular and functional analysis of the mitochondrial respiratory chain.** (A) TCGA analysis showing expression values of respiratory chain complexes as a function of low, normal, and high expression of MICU1, MICU2, and the MICU2/MICU1 ratio. (B) Boxplot representing the average expression of genes associated with the assembly of mitochondrial respiratory chain complexes I, II, III, and IV based on the status MICU1 and MICU2/MICU1 ratio in primary colon tumor samples from the TCGA-COAD data set. (C) The expression levels of MCU, MICU1, and MICU2 were determined in the Control KO and MICU2 KO cells treated with the inhibitors rotenone and oligomycin using RT-qPCR. The data are presented as the mean ± standard error of the mean of 4 independent experiments. On all plots, $^*p < 0.05$, $^{**}p < 0.01$, and $^{***}p < 0.001$. The data underlying the graphs shown in the figure can be found in S1 and S2 Datas.
(TIF)

**S5 Fig. The impact of MICU2 deletion on fatty acid metabolism and beta-oxidation activities.** (A) The expression of MCU, MICU1, and MICU2 were determined in the Control and MICU2 KO cell lines treated with the inhibitors etomoxir (left panel) and trimetazidine (right panel) using RT-qPCR. The data are presented as the mean ± standard error of the mean of 3 or 4 independent experiments. (B) Efficacy of V-9302 (ASCT2 inhibitor) (left panel) and telaglenastat (glutaminase inhibitor) (right panel) treatments (1–10 μm) after 48 h on the Control and MICU2 KO cell lines ($n = 5$, ANOVA followed by Dunn's multiple comparisons test, $^*p < 0.05$). (C) qRT-PCR data showing fold changes in mRNA levels of glycolysis regulator proteins in the Control and MICU2 KO cell lines ($n = 4–7$, ANOVA followed by Dunnett's multiple comparisons test, $^*p < 0.05$ and $^{**}p < 0.01$). (D) RT-qPCR data showing fold changes in mRNA levels of pyruvate regulator proteins in the Control and MICU2 KO cell lines ($n = 3–5$, ANOVA followed by Dunnett's multiple comparisons test, $^{**}p < 0.01$). (E) RT-qPCR data showing fold changes in mRNA levels of pentose phosphates proteins in the Control KO and MICU2 KO cell lines ($N = 5$, ANOVA followed by Dunnett's multiple comparisons test, $^*p < 0.05$ and $^{***}p < 0.001$). The data underlying the graphs shown in the figure can be found in S1 Data.
(TIF)

**S6 Fig. Expression of MICU2 related to genes involved in glycolysis, hypoxia, mitochondrial respiration, and the TCA cycle.** (A) The expression of MCU, MICU1, and MICU2 was determined in the Control and MICU2 KO cell lines treated with the inhibitor UK5099 using RT-qPCR. The data are presented as the mean ± standard error of the mean of 4 independent experiments. (B) Heatmap representing the expression of genes associated with glycolysis, hypoxia, mitochondrial respiration, and the TCA cycle as a function of the MICU2/MICU1 status of primary tumor samples from the TCGA-COAD data set. (C) Boxplot representing the mean expression of genes associated with glycolysis, hypoxia, mitochondrial respiration, and the TCA cycle according to the MICU1 and MICU2/MICU1 ratio status of primary tumor samples of the TCGA-COAD data set. (D) Scatter plot representing the correlation of the average expression of genes associated with glycolysis, hypoxia, mitochondrial respiration, and the TCA cycles as a function of the expression of MICU1, MICU2, or the MICU2/MICU1 ratio in a panel of 57 CRC cell lines from the RNA-Seq data set of the CCLE (R is the Pearson correlation coefficient). (E) Boxplots representing the average expression of genes associated with glycolysis, hypoxia, mitochondrial respiration, and the TCA cycle in a panel of 57 CRC cell lines from the RNA-Seq data set of the CCLE classified as low or high expression of MICU1, MICU2, or the MICU2/MICU1 ratio as a function of the median expression of MICU1, MICU2, or the MICU2/MICU1 ratio. On all plots, $^*p < 0.05$, $^{**}p < 0.01$, and $^{***}p < 0.001$. The data underlying the graphs shown in the figure can be found in S1 and S2 Datas.
(TIF)

**S7 Fig. Graphical Abstract.** Graphical summary of the main study results.
(TIF)

**S1 Data. Raw data supporting Figs 1–6 and Supplementary Figs 1–6.**
(XLSX)

**S2 Data. Data supporting the GSEA analysis.**
(ZIP)

**S3 Data. Differential expression analysis of TCGA-COAD tumors in function of MICU2 expression.**
(ZIP)

**S4 Data. Differential expression analysis of TCGA-COAD tumors in function of MICU2/ MICU1 expression.**
(ZIP)

**S1 Table. List of primers for real-time polymerase chain reaction.**
(DOCX)

**S2 Table. List of key reagents and resources.**
(DOCX)

**S1 Raw Images. Western blot sets used in the article.**
(PDF)

# Acknowledgments

The authors acknowledge H2P2 platform for technical assistance for immunohistochemistry analysis (Nicolas Mouchet, Anthony Sébillot, Gevorg Ghukasyan–Université Rennes 1) and PHENOMIN-TAAM-UPS44, CIPA (Alain Le Pape and Sharuja Natkunarajah - Centre d'Imagerie du Petit Animal, part of MOV2ING platform, Orléans) for in vivo models. Thanks to

Frederic Picou, member of the N2CoX and Jacques Dupuy, INRAe Toxalim, for their help. Thanks to Agathe Brugoux, Oceane Pertegaz, Beatrice Genova, Cyrille Guimaraes, and Michelle Pinault members of the N2COx, for their technical help.

Inserm UMR 1069 is leader of Cancéropôle Grand-Ouest 3MC network (Marine Molecules, Metabolism and Cancer), member of Région Centre–Val de Loire thematic research consortium RTR MOTIVHEALTH (Molecular and Technological Innovation for Health), and member of CNRS research group APPICOM (GDR2082 - Integrative Approach for a multiscale functional analysis of membrane proteins). Alison Robert, Maxime Guéguinou, Arnaud Chevrollier, Naig Gueguen, Jean-François Dumas, and William Raoul are also members of Meetochondrie French network.

## Author Contributions

**Conceptualization:** Maxime Guéguinou.

**Data curation:** Alison Robert, David Crottès, Jérôme Bourgeais, Naig Gueguen, Arnaud Chevrollier, Isabelle Domingo.

**Formal analysis:** Alison Robert, David Crottès, Jérôme Bourgeais, Naig Gueguen, Arnaud Chevrollier, Jean-François Dumas, Stéphane Servais, Isabelle Domingo, Stéphanie Chadet, Julien Sobilo, Olivier Hérault, Thierry Lecomte, Christophe Vandier, William Raoul, Maxime Guéguinou.

**Funding acquisition:** Olivier Hérault, Thierry Lecomte, Christophe Vandier, William Raoul, Maxime Guéguinou.

**Investigation:** Alison Robert, David Crottès, Jérôme Bourgeais, Naig Gueguen, Arnaud Chevrollier, Stéphane Servais, Isabelle Domingo, Stéphanie Chadet, Julien Sobilo, Christophe Vandier, Maxime Guéguinou.

**Methodology:** Alison Robert, David Crottès, Jérôme Bourgeais, Naig Gueguen, Arnaud Chevrollier, Jean-François Dumas, Stéphane Servais, Julien Sobilo, Olivier Hérault, Thierry Lecomte, William Raoul, Maxime Guéguinou.

**Project administration:** William Raoul, Maxime Guéguinou.

**Supervision:** Olivier Hérault, Thierry Lecomte, Christophe Vandier, William Raoul, Maxime Guéguinou.

**Writing – original draft:** Alison Robert, David Crottès, William Raoul, Maxime Guéguinou.

**Writing – review & editing:** Alison Robert, David Crottès, Jérôme Bourgeais, Naig Gueguen, Arnaud Chevrollier, Jean-François Dumas, Stéphane Servais, Stéphanie Chadet, Julien Sobilo, Olivier Hérault, Thierry Lecomte, Christophe Vandier, William Raoul, Maxime Guéguinou.

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
