## [Editor Report · Decision Letter 0]

25 Nov 2023

Dear Dr Gueguinou, 

Thank you for submitting your manuscript entitled "Encoding of the colorectal cancer metabolic program through MICU2" for consideration as a Research Article by PLOS Biology.

Your manuscript has now been evaluated by the PLOS Biology editorial staff as well as by an academic editor with relevant expertise and I am writing to let you know that we would like to send your submission out for external peer review.

Once your full submission is complete, your paper will undergo a series of checks in preparation for peer review. After your manuscript has passed the checks it will be sent out for review. To provide the metadata for your submission, please Login to Editorial Manager (https://www.editorialmanager.com/pbiology) within two working days, i.e. by Nov 29 2023 11:59PM.

Kind regards,

Ines

--

Ines Alvarez-Garcia, PhD

Senior Editor

PLOS Biology

---

## [Decision Letter · Decision Letter 1]

29 Feb 2024

Dear Dr Gueguinou,

Thank you for your patience while your manuscript entitled "Encoding of the colorectal cancer metabolic program through MICU2" was peer-reviewed at PLOS Biology. Please also accept my sincere apologies for the time it has taken us to provide you with a decision. The manuscript has now been evaluated by the PLOS Biology editors, an Academic Editor with relevant expertise, and by two independent reviewers.

As you will see, the reviewers find the conclusions interesting, but they also raise several concerns that would need to be addressed before we can consider the manuscript for publication. Reviewer 1 thinks you should show if MICU2 depletion alters the size and formation of MCU, that you need to use other cell lines to confirm the findings, add quantification to some experiments, and improve the methods and some of the figures. Reviewer 2 also thinks it’s important to analyse the effect of the lack of MICU2 in metastatic spread in vivo, and asks for several clarifications and missing information in the text.

In light of the reviews, we would like to invite you to revise the work to thoroughly address the reviewers' reports. Given the extent of revision needed, we cannot make a decision about publication until we have seen the revised manuscript and your response to the reviewers' comments. Your revised manuscript is likely to be sent for further evaluation by all or a subset of the reviewers.

**IMPORTANT - SUBMITTING YOUR REVISION**

3. Resubmission Checklist

a) *PLOS Data Policy*

b) *Published Peer Review*

Sincerely,

Ines

--

Ines Alvarez-Garcia, PhD

Senior Editor

PLOS Biology

Reviewers' comments

Rev. 1:

The manuscript concerns the mitochondrial consequences of loss of MICU2, part of the MCU. They investigate how this changes metabolism of the cell and some aspects of tumor phenotype. They show that loss of this protein leads to changes in mitochondrial fusion/fission, respiratory supercomplex expression, respiration (which impacts on central carbon metabolism more broadly) and some malignant phenotypes, such as invasion. It is an interesting study, which would benefit from a clearer link to tumors themselves - while small mRNA changes are observed, do these translate into protein changes, on which this manuscript is based?

Other comments:

1. The formation of the multiprotein complex MCU has been shown to depend on association of MICU1 and MICU2 as a heterodimer (Mallilankaraman et al 2012; Csordas et al 2013 and others). Given that the authors have shown that they are able to run native gels for supercomplexes, it would be important for this manuscript to show whether depletion of MICU2 alters the size and formation of MCU, given that MICU1 protein expression is unchanged (Figure 1D).

2. Figure 1C compares MICU1, 2 and MCU expression in cancer cell lines against a normal cell line. It would be best to include more than one representative normal cell line, as a single model is not enough to draw a conclusion regarding differences between normal and cancer. Additionally, it isn't clear that MICU1 or MCU are higher in the cancer cell lines than the normal - loading is not consistent between normal and the cancer cells, and given that western blotting is not fully quantitative, best practice would suggest quantification should really only be done on blots with equal loading.

3. In Figure 2C, this reviewer is not aware that it is possible to separate G0 from G1, or G2 from M through PI fluorescence by flow cytometry. If this is the case, could the authors outline in the materials and methods how this was performed, and if not, could they collapse the categories into broader cell cycle phases.

4. The results arising from Ki67 staining of tumors shown in Figure 2F were unexpected, in that the two knockdown constructs appeared to behave in different ways. Additionally, the pictures appear to be at different magnifications, despite the scale bar suggesting otherwise. The control KO punctae for Ki67 nuclear staining are significantly smaller than those for the MICU #B KO picture. In engraftment delay is indeed the reason for a change in proliferation, then the authors would need to take tumors of similar size (independent of engraftment) and examine these to compare like with like.

5. Figure 4B and S4A and B. The authors suggest that their analysis of the TCGA-COAD data shows that MICU1 high tumors have higher expression of genes associated with the assembly of the complexes, and go onto show changes in some supercomplex formation. Could the authors consider also showing protein expression of the assembly factors to support the RNA changes observed in tumor samples?

6. At the end before the section of the results describing Figure 4, the authors state that 'overall, we demonstrated that MICU2, unlike MICU1, controls the complex biogenesis…' While in this manuscript they do show that depletion of MICU2 alters supercomplex expression in cells, they do not show that depletion of MICU1 does not. They would need to refer to previously published work here to be able to make that comparison.

7. In Figure 5D the authors present mRNA for the PLINs. The stability of PLIN2 expression is controlled post-translationally, so it would be more informative to show protein expression.

8. In Figure 6, the authors note that tumor oxygenation is lower in MICU2 KO cell clone A-derived tumors (6G). This is unexpected, given that for a given tumor size, these cells use less oxygen, and therefore the diffusion gradients across the tumor are likely to be less steep, leading to relatively higher oxygenation - similar to that seen after atovaquone treatment in a number of other studies. The increased glycolysis cannot be used as an indicator of a preference for growing in hypoxia, and this preference will not change the environment - the environment is create due to a combination of supply and demand for oxygen. As the authors have shown that there is increased hypoxia, and that the tumor cells use less oxygen, the supply must be significantly less in these tumors - this will be important to look for, or an alternative explanation.

Minor comments:

1. In Figure 2C, the colors are a bit hard to differentiate between, particularly the orange plot. Could the authors add the percentage cells in each phase of the cell cycle to the plot?

Rev. 2:

The paper by Robert et al. explores the role of MICU2, and EF-hand containing regulator of the mitochondrial calcium uniporter MCU in the development and metabolic reprogramming of colon carcinoma. The authors used a vast variety of techniques in combination with bioiformatics in order to prove their hypotheses. The work is performed carefully, however a number of points require clarification.

Introduction:

Please specify which isoform of VDAC has been linked to calcium transport across the OMM.

Please specify whether Garg et al reported activation of MCU by both MICU1 and MICU2. If yes, it is difficult to understand why expression of MICU1 and MICU2 are respectively down- and upregulated with stages of CRC, as cited in the results section. In addition, upregulation of MICU2 would counteract the Warburg effect/lactate production according to the graphical abstract based on the results. Is it known from the literature if CRC cells tend to suppress the Warburg effect with stage advancement and metastatic spread? MICU2 KO cells, in which more lactate is produced, gives rise to smaller tumors. This is somewhat unexpected, please give possible explanation. Is it possible that more lactate does not aid tumor growth because the experiments were performed in NOD-SCID mice where lactate probably does not remodel the tumor microenvironment?

The author write: "So far, the contribution of MICU2 to the regulation of mitoCa2+ signaling, bioenergetics, metabolic reprogramming, mitochondrial dynamics, and CRC development have not been addressed."

There are several studies addressing the role of MICU2 in the regulation of mitochondrial calcium signaling, dynamics and metabolic reprograming. Please modify the sentence accordingly and cite those studies.

Results:

In addition to reference 12, please cite the earlier papers where the effects of MICU1 and MICU2 have been studied on calcium flux into the mitochondrial matrix (works by the groups of Mootha, Hajnoczky, Rizzuto).

"Significantly, MCU, MICU1 and MCUb were negatively expressed while MICU2 was positively expressed". How can be a protein negatively expressed? Maybe the authors refer to a lower expression compared to healthy tissues"

Figure 1C: The authors show an increased expression of MCU, MICU1 and MICU2 with increasing aggressiveness. However, at the beginning of the result section they write "In particular, using The Cancer Genome Atlas Colon Adenocarcinoma (TCGA-COAD) dataset, the authors observed that the expression of MICU1 and MICU2 are respectively down- and upregulated with stages". Please explain.

BAPTA-AM seems to decrease viability regardless of the presence of MICU2 and the conclusion of the authors from this experiment is not clear.

Since MICU2 KO lines show decreased migration, on the other hand MICU2 is highly expressed in metastatic cells, it would be crucial to understand the effect of the lack of MICU2 in metastatic spread in vivo. This would significantly strengthen the paper and link the observations in preclinical models with clinical data.

Please write out PRPS1.

Discussion:

Please discuss whether the known chemical MICU/MCU modulators might be useful to decrease tumor growth in light of your findings.

Materials and methods:

The methodologies used to obtain Figures 2A and 6J are not sufficiently described. Please specify the program used to perform these analyses.

---

## [Decision Letter · Decision Letter 2]

16 Aug 2024

Dear Dr Gueguinou,

Thank you for your patience while we considered your revised manuscript "Encoding of the colorectal cancer metabolic program through MICU2" for publication as a Research Article at PLOS Biology. Please note that I am currently handling your manuscript as my colleague Ines Alvarez-Garcia is away from the office this week. This revised version of your manuscript has been evaluated by the PLOS Biology editors, the Academic Editor and the original reviewers.

Based on the reviews, I am pleased to say that we are likely to accept this manuscript for publication, provided you satisfactorily address the following data and other policy-related requests that I have provided below (A-H):

(A)We would like to suggest the following minor modification to the title:

“Upregulation of MICU2 facilitates cell proliferation and metabolic reprogramming during colorectal cancer development”

(B) Please include the full name of the IACUC/ethics committee that reviewed and approved the animal care and use protocol/permit/project license in the Methods section. Please also include an approval number.

(C) You may be aware of the PLOS Data Policy, which requires that all data be made available without restriction: http://journals.plos.org/plosbiology/s/data-availability. For more information, please also see this editorial: http://dx.doi.org/10.1371/journal.pbio.1001797

-Supplementary files (e.g., excel). Please ensure that all data files are uploaded as 'Supporting Information' and are invariably referred to (in the manuscript, figure legends, and the Description field when uploading your files) using the following format verbatim: S1 Data, S2 Data, etc. Multiple panels of a single or even several figures can be included as multiple sheets in one excel file that is saved using exactly the following convention: S1_Data.xlsx (using an underscore).

-Deposition in a publicly available repository. Please also provide the accession code or a reviewer link so that we may view your data before publication. 

Figure 1B-E, 2A-F, 3A-D, 3F-H, 4B-F, 5A, 5C-K, 6B-K, S1A-E, S2A-E, S2G, S3A-F, S4A-C, S5A-E, S6A-C

(D) Please also ensure that each of the relevant figure legends in your manuscript include information on *WHERE THE UNDERLYING DATA CAN BE FOUND*, and ensure your supplemental data file/s has a legend.

(E) We require the original, uncropped and minimally adjusted images supporting all blot and gel results reported in the following Figures:

Figure 1C-E, 3D, 4C

We will require these files before a manuscript can be accepted so please prepare and upload them now. Please carefully read our guidelines for how to prepare and upload this data: https://journals.plos.org/plosbiology/s/figures#loc-blot-and-gel-reporting-requirements

(F) Please ensure that your Data Statement in the submission system accurately describes where your data can be found and is in final format, as it will be published as written there. 

(G) Per journal policy, if you have generated any custom code during the course of this investigation, please make it available without restrictions. Please ensure that the code is sufficiently well documented and reusable, and that your Data Statement in the Editorial Manager submission system accurately describes where your code can be found. 

(H) Please note that per journal policy, the model system/species studied should be clearly stated in the abstract of your manuscript. 

We expect to receive your revised manuscript within two weeks. 

*Published Peer Review History*

*Press*

Kind regards,

Richard 

Richard Hodge, PhD

rhodge@plos.org

On behalf of:

Ines Alvarez-Garcia, PhD

Reviewer remarks:

Reviewer #1: The authors have comprehensively attempted to address all my comments, and where reagents are unadequate this is clear, and unfortunately cannot be remedied.

Reviewer #2: The authors addressed all my concerns, also by performing new experiments.

---

## [Editor Report · Decision Letter 3]

20 Sep 2024

Dear Dr Gueguinou,

Thank you for the submission of your revised Research Article entitled "MICU2 upregulation enhances tumor aggressiveness and metabolic reprogramming during colorectal cancer development." for publication in PLOS Biology. On behalf of my colleagues and the Academic Editor, Heather Christofk, I am delighted to let you know that we can in principle accept your manuscript for publication, provided you address any remaining formatting and reporting issues. These will be detailed in an email you should receive within 2-3 business days from our colleagues in the journal operations team; no action is required from you until then. Please note that we will not be able to formally accept your manuscript and schedule it for publication until you have completed any requested changes.

PRESS

Sincerely, 

Ines

--

Ines Alvarez-Garcia, PhD

Senior Editor

PLOS Biology
